

# Constructing Fast and Representative Analytical Models of Wind Turbine Main-Bearings

James Stirling [1], Edward Hart [2], and Abbas Kazemi Amiri [2]

[1]Wind and Marine Energy Systems CDT, EEE, University of Strathclyde, Glasgow, UK
[2]Wind Energy and Control Centre, EEE, University of Strathclyde, Glasgow, UK

**Correspondence:** James Stirling, Wind and Marine Energy Systems CDT, EEE, University of Strathclyde, Glasgow, UK
(j.stirling@strath.ac.uk)

**Abstract.** This paper considers the modelling of wind turbine main bearings using analytical models. The validity of simplified analytical representations used in existing work is explored by comparing main bearing force reactions with those obtained from higher fidelity 3D finite element models. Results indicate that there is good agreement between the analytical and 3D models in the case of a non moment-reacting case (such as for a spherical roller bearing), but, the same does not hold in the moment reacting case (such as for tapered roller bearings). Therefore, a new analytical model is developed in which moment reactions at the main bearing are captured through the addition of torsional springs. This latter model is shown to improve the agreement between analytical and 3D models in the moment reacting case. The new analytical model is then used to investigate load characteristics, in terms of forces and moments, for this type of main bearing across different operating points and wind conditions.

## 1 Introduction

Wind energy provides an important and growing contribution to the European energy market, with 205GW installed as of 2019 - accounting for 15% of consumed electricity (Wind Europe, 2020). As part of this growth, more wind farms are being planned and constructed offshore to take advantage of higher wind speeds and more vailable construction space (Junginger et al., 2004). Recent trends show dramatic falls in the cost of offshore wind, as been mirrored in the UK's contract for difference auctions which have seen prices drop to £57.50/MWh (UK Government, 2017) and even lower.

With turbines moving further offshore and a need to bring costs down means that reducing operation and maintenance costs, which can be as high as 35% of the total lifetime costs of a project, is becoming increasingly important for wind farm operators (Sinha and Steel, 2015). This in turn effects technology design and selection and puts pressure on original equipment manufacturers (OEMs) and operators to improve turbine reliability. As such reliability and failure rate considerations have received much attention in the literature (Walford, 2006; Tavner et al., 2007; Wilkinson, 2011).

One turbine component with relatively high failures rates and associated downtime is the main bearing (MB). MBs are becoming recognised as an important component for which failures need to be better understood and reliability improved (Keller et al., 2016; Hart et al., 2019a). MB failure rates have been reported as being up to 30% (Hart et al., 2019b) across a 20 year lifetime, with some wind farms having reported MBs failing in less than 6 years (Sethuraman et al., 2015).





Preventing premature failures of main-bearings would therefore be an important contribution to reducing operating costs of wind farms. As part of analyses which try to understand the loading conditions of MBs in wind turbines (in order to better understand their operational conditions and load characteristsics), detailed model-based investigations are required. Work of this type exists in the literature (Hart et al., 2019b) in which analytical models are used to consider MB loading. This paper considers the validity of simplified analytical drivetrain representations of the type used in these load studies by comparisons

with higher fidelity 3D finite element models.

    Section 2 provides a description of the previous work undertaken in this area. Section 3 then introduces the higher fidelity 3D models which will be used to validate the analytical models before presenting the results of the comparison. In section 4 the analytical model is adapted to include moment reactions at the MB, before comparing the models again. Section 5 applies the new analytical model to study load behaviours for this bearing type. Finally, Section 6 presents the conclusions of this work.

## 2   Previous Work

A proper understanding of main-bearing loading requires full consideration of the complex load environment with which the bearing is interacting. This work expands from work completed previously (Hart et al., 2019b) in which hub loading time histories were generated using multi-body aero-elastic software and injected into simplified 2 dimensional models of realistic MB set-ups to determine MB operational loading.

A set of 3-dimensional turbulent wind fields were generated in DNV-GL's Bladed software using a Kaimal spectrum in accordance with IEC standards and six different wind fields were created for every combination of the selected wind parameters as required for design certification (IEC, 2005). The three parameters focused on were hub-height mean wind speeds (10, 12, 16, 20m/s), turbulence intensity (low, medium and high as specified by IEC (2005)) and shear profile (power law shear exponents of 0.2 and 0.6) resulting in a total of 144 wind profiles spanning a significant range of typical operational conditions. The 6

wind files associated with each combination of the parameters will be referred to as common parameter load sets (CPLS). Bladed was then used to run each wind file through fully aero-elastic multi-body simulations of a 2MW wind turbine and the hub loading time series extracted.

    Simple engineering drawings were provided by Onyx Insight for the study undertaken in Hart et al. (2019b) which provide the dimensions of various MB set-ups and included the gearbox connections as spring stiffness values. Three analytical models

were then created including a single main-bearing (SMB) system and two double main-bearing (DMB) systems. The hub loading time series across the full range of wind files were then injected into the models and the bearing reaction forces extracted. The analytical model for the SMB drivetrain configuration is displayed in Figure 1 and will form the focus of this paper.

    The equation system for the SMB drivetrain set-up is statically determinate and can be solved by balancing the moments

about the gearbox giving:

$$F = \frac{M + (L_1 + L_2)B}{L_2} \tag{1}$$



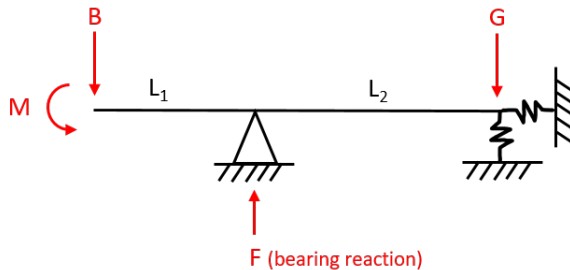

**Figure 1.** 2D analytic model for single main-bearing set-up (Hart et al., 2019b).

It is important to note that the overall model consists of two of the type shown in Figure 1, with one in the horizontal and one in the vertical plane, with the resultant force being a vector combination of the two. $B$ and $M$ represent the force and moment loads at the hub, $L_1$ and $L_2$ are the distances between the hub and MB and MB and gearbox, respectively, and G represents the gearbox mass in the vertical plane and is zero in the horizontal plane. $F$ is the bearing support reaction force. Findings demonstrated greatly varying mean and peak loads, as well as load ratios, between the different drivetrain configurations and high sensitivities to wind field characteristics.

While models and results in Hart et al. (2019b) are promising, the utilised models are simple, and hence come with limitations. The bearings are modelled as single point fixed supports, meaning all loads are reacted by a reaction force at the MB with no moment reaction present. The two most common bearings used for WT MBs are spherical roller bearings (SRB) which cannot react moment loads and tapered roller bearings (TRB) which can react both forces and moments (Yagi, 2004; Smalley, 2015; Hart et al., 2019a). Therefore, the validity of existing models for different bearing types should be considered. This validity is the focus of the current work.

## 3 Comparison of Analytical and FEA models

In order to asses the effectiveness of the simple analytical models used thus far, two finite element (FE) models of the SMB system were created in ANSYS; with one designed to behave like an SRB, and the other to behave like a TRB, as described below. The models were subjected to the same hub loading as the analytical models, outlined in the previous section, with bearing support reaction forces outputted and compared with those from the analytical model. Both FE models share dimensions with the SMB analytical model. The FE models themselves still remain relatively simple, with relevant behaviours captured without the modelling of individual rolling elements as described below.

**SRB FE Model** - The SRB FE model was created with 3 separate bodies; the shaft, the bearing and the bearing housing. A fixed support was added to the base of the bearing housing to represent the connection to the bed plate and the connections between low speed shaft and gearbox was modelled by spring connections horizontally and vertically with stiffness values determined by Romax Technology software. The bearing was connected to the body with a bonded connection and the bearing





to bearing housing connection modelled as a deformable spherical joint. This type of connection allows the transfer of force loads from the shaft to the bearing and housing but will not react moments, emulating SRB behaviour. A sliced view of the bearing, housing and shaft can be seen in Figure 2a side-by-side with SRB elements overlaid on the same image to demonstrate the interface type being represented. The mesh was sized to have larger element sizes across the shaft with smaller elements around the bearing and bearing housing to increase accuracy at the contact regions.

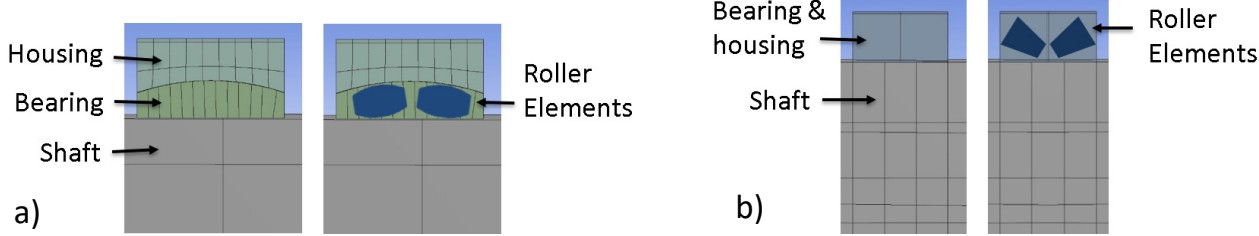

**Figure 2.** (a) A split view of the SRB FE model displaying the geometries of the bearing and housing. (b) A split view of the TRB FE model displaying the geometries of the bearing and housing.

**TRB FE Model** - The TRB FE model was created with two separate bodies; the shaft and the bearing/bearing housing. The bearing and bearing housing were modeled as one piece of material with a bonded connection to the shaft to emulate the moment reaction properties of a TRB and the preloading of rollers. A cross section of the model is displayed in Figure 2b. The base of the main-bearing housing was modelled with a fixed support to represent the connection to the main bed plate and the gearbox was again modelled by body-to-ground horizontal and vertical spring connections with the same stiffness properties

as the SRB model.

Plots of root mean squared error (RMSE) comparison results between the analytical and two FE models are shown in Figures 3 and 5, along with example time series plots of the bearing unit reaction forces in Figures 4 and 6. The RMSE plots present the mean and standard deviations from within each CPLS (which each capture results from 6 wind files with parameters in common) with respect to mean wind speed, turbulence intensity and shear profile. Note that mean wind speeds are staggered

for clarity.

Figure 3 displays RMSE results between the analytical model and the SRB FE model in the horizontal and vertical planes. The accuracy of the analytical model in the horizontal axis appears to be sensitive to mean wind speeds and shear exponents, decreasing as their values increase. The RMSE results for the bearing reaction force in the vertical axis are less differentiated by the varying wind parameters than in the horizontal axis. There is, however, still a slight trend of decreasing accuracy as

hub height mean wind speed and shear exponents are increased. To put these results into context, the mean percentage error between resultant force magnitudes for the two across all wind files is 9.27%, with a mean correlation coefficient of 0.957. These results indicate that the analytical model does in fact give good results across all tested wind profiles in both planes when compared with 3D model outputs. This conclusion is reinforced when one considers time series of these loads, with a examples shown in Figure 4.

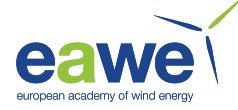


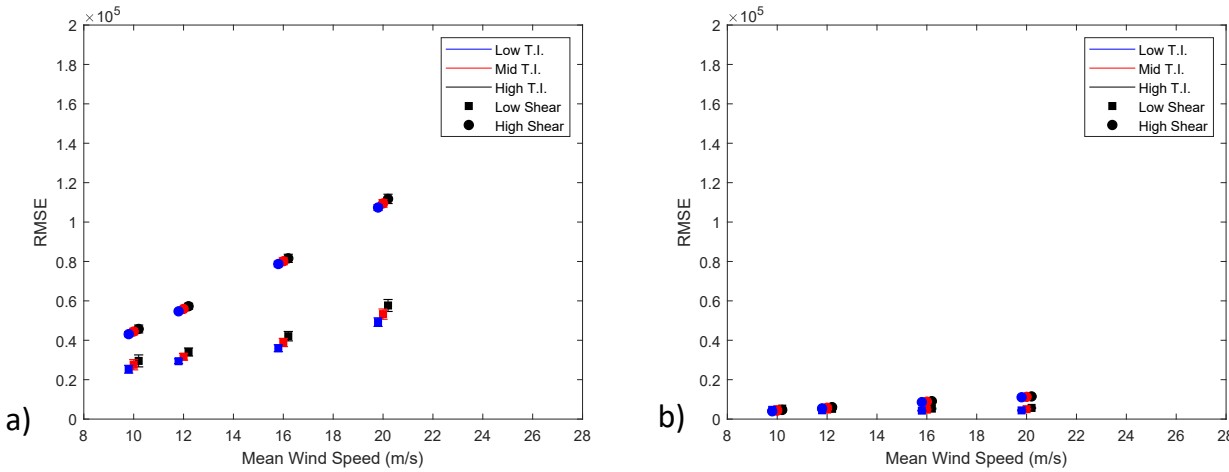

**Figure 3.** (a) RMSE between reaction forces from the analytical and SRB FE model in the horizontal plane. The mean and standard deviations within each CPLS are plotted, staggered about mean wind speed for clarity. (b) RMSE reaction force results between the analytical and SRB FE model in the vertical plane. The mean and standard deviations within each CPLS are plotted, staggered about mean wind speed for clarity.

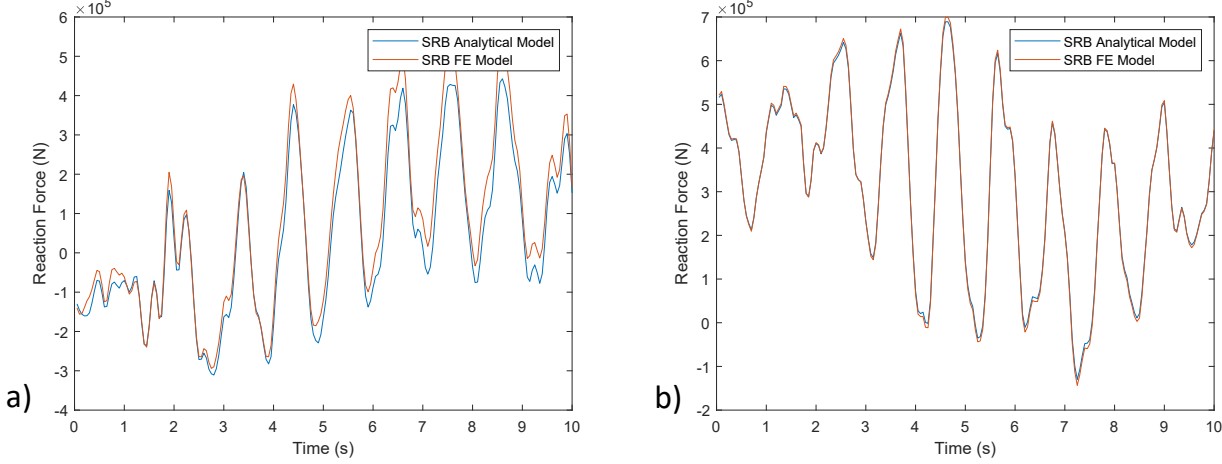

**Figure 4.** (a) Example time series of reaction force results in the horizontal plane from the analytical and SRB FE models. (b) Example time series of reaction force results in the vertical plane from analytical and SRB FE models.

The analytical model reaction force results were then compared with the TRB FE model, with the results displayed in Figure 5. The analytical model shows a trend of decreasing accuracy with increasing wind speed and shear in the horizontal plane. Compared to the previous results, error values can be seen to have significantly increased. Increasing mean wind speeds for low shear does not appear to effect the accuracy of the model in the vertical plane, but, sensitivities to turbulence intensity are evident. In contrast, the high shear exponent results in the vertical plane decrease in accuracy significantly with increasing mean



wind speeds. The mean percentage error and correlation coefficient were again considered between resultant force magnitudes
across all wind files. The mean error was found to be 30.07% and a mean correlation coefficient of 0.782 was calculated,
showing that the analytical model is noticeably less accurate in the TRB moment reacting case. This conclusion is again
reinforced by time series of model outputs, examples of which are shown in Figure 6.

The comparison results between the analytical and FE models suggests that the analytical model is generating valid force
outputs in the SRB case. However, the results also show that the analytical model cannot accurately represent force reactions
for a TRB system. This motivates the derivation of a new analytical model in order to try and emulate the positive results seen
in the SRB for a TRB setup. Such a model is developed in the following section.

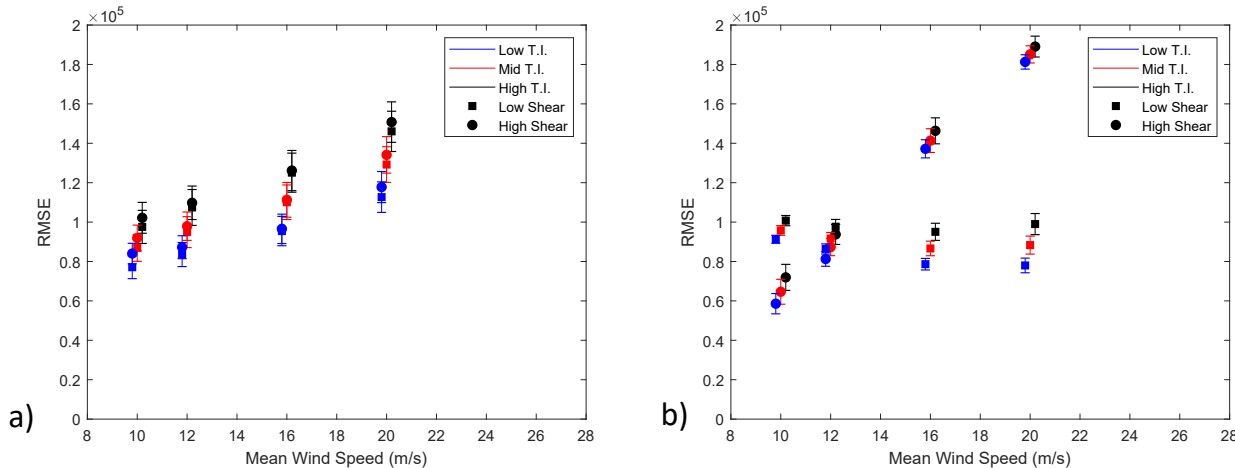

**Figure 5.** (a) RMSE between reaction forces from the analytical and TRB FE model in the horizontal plane. The mean and standard deviations within each CPLS are plotted, staggered about mean wind speed for clarity. (b) RMSE reaction force results between the analytical and TRB FE model in the vertical plane. The mean and standard deviations within each CPLS are plotted, staggered about mean wind speed for clarity.

## 4  Adapting the analytical model for moment reactions

In order to facilitate moment reactions at the MB, torsional springs were added to the fixed bearing support in both planes of
the analytical model. Thus, a new analytical model was created, displayed in Figure 7 (a). The set of equations for the new
analytical model are statically indeterminate and so the model must be decoupled to find a solution (Hibbeler, 2011; Leet
et al., 2011). The model was first simplified by moving the location of the force applied by the rotor mass, $B$, and associated
overturning moment, $M$, to be positioned at the bearing support mount as shown in Figure 7 (b). The model was then decoupled
into two deflection models; one which has the rotor weight and overturning moment acting on the structure (Figure 8 (a)) and
one which has the reaction moment from the torsional spring acting on the structure (Figure 8 (b)).





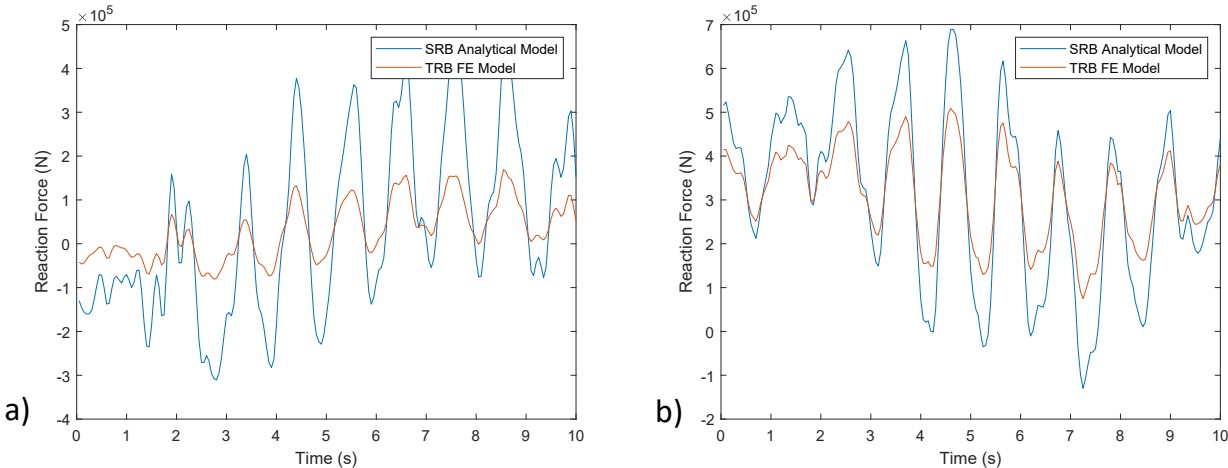

**Figure 6.** (a) Example time series of reaction force results in the horizontal plane from the analytical and TRB FE models. (b) Example time series of reaction force results in the vertical plane from analytical and TRB FE models.

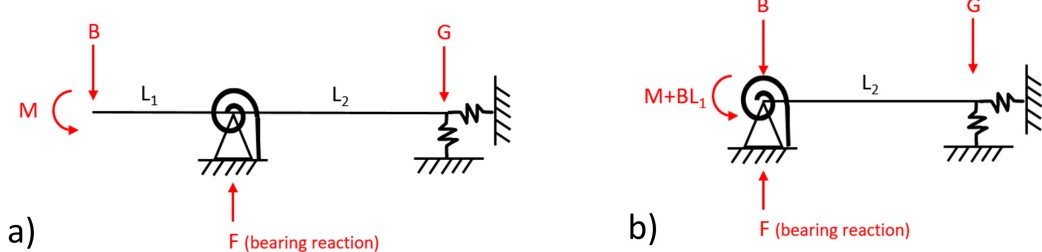

**Figure 7.** (a) 2D analytical model for single main-bearing set-up with torsional spring to include moment reactions. (b) Simplified 2D analytical model with torsional spring.

The two deflection models can then be decoupled again to show the two mechanisms causing deflection in the shaft; bending of the beam due to the applied moment, and rotation about the main-bearing support due to spring support (gearbox) compression/extension. As the deflection mechanisms and equation derivation process is similar for the overturning moment and spring reaction moment on the system, only the equations and deflection mechanisms for the overturning moment is presented here.

The two deflection mechanisms for the decoupled model with overturning moment and rotor weight is shown in Figure 9.

Calculating $\theta_{11}$ as seen in Figure 9 can be done by utilising the beam deflection formula shown in Equation 2 (Popov, 1990).

$$\theta_{11} = \frac{(M + BL_1)L_2}{3EI} \qquad (2)$$





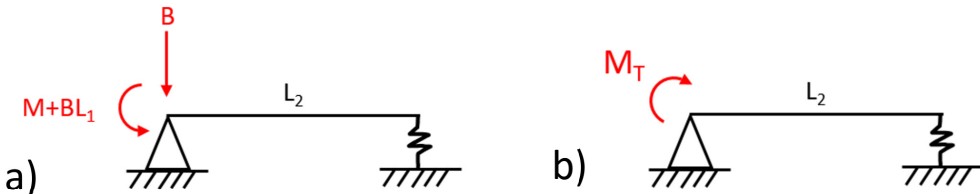

**Figure 8.** (a) Deflection model 1 (rotor weight and overturning moment). (b) Deflection model 2 (torsional spring reaction force).

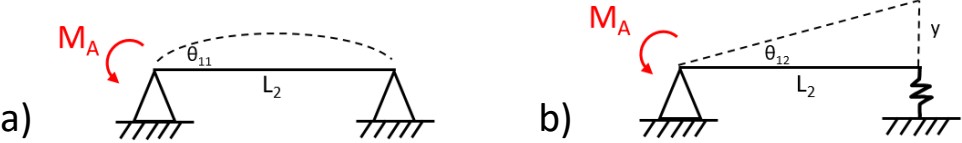

**Figure 9.** Deflection mechanisms for deflection model 1 under some applied moment $M_A$.

Calculating $\theta_{12}$ is not as straightforward as the compression/extension length, $y$, of the spring must first be found. For a loaded spring with stiffness, $k$, the distance stretched or compressed, $y$, is equal to the reaction force divided by the stiffness.

$$y = \frac{R_B}{k}, \tag{3}$$

Trigonometrically, the deflection angle is then,

$$\tan\theta_{12} = \frac{y}{L_2}, \tag{4}$$

and a small-angle approximation simplifies the equation to,

$$\theta_{12} = \frac{y}{L_2}, \tag{5}$$

and so subbing in Equation 3 for $y$ gives,

$$\theta_{12} = \frac{R_B}{KL_2}. \tag{6}$$

The second set of deflection equations with respect to the reaction moment of the torsional spring on the shaft are calculated using the same method, with the angles of rotation labelled $\theta_{21}$ and $\theta_{22}$ and taking values of,

$$\theta_{21} = \frac{M_T L_2}{3EI}, \tag{7}$$

and,

$$\theta_{22} = \frac{R_T}{KL_2}. \tag{8}$$

The rotation of the torsional spring, $\theta_{TS}$, is given by,

$$\theta_{TS} = \frac{-M_T}{K_R}, \tag{9}$$



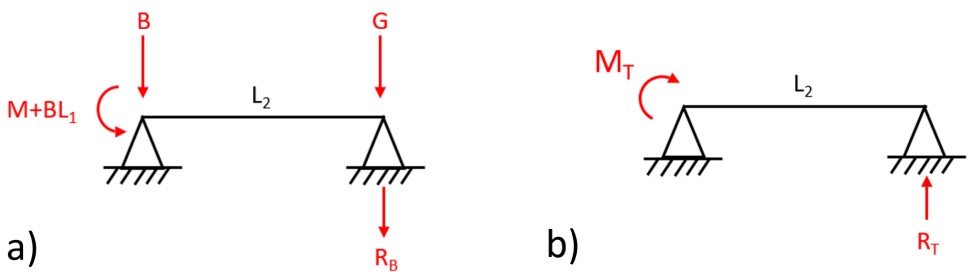

**Figure 10.** (a) Force balance corresponding to deflection model 1. (b) Force balance corresponding to deflection model 2.

where $K_R$ is the stiffness of the spring and $M_T$ the reaction moment. The rotation of the torsional spring is also equal to the
sum of all deflection angles, with positive and negative signs indicating direction,

$$\theta_{TS} = -\theta_{11} - \theta_{12} + \theta_{21} + \theta_{22}. \qquad (10)$$

The reaction forces $R_B$ and $R_T$ are still unknowns and the above equation cannot be solved until the forces are balanced on
the decoupled models. Balancing the moments about the bearing support in Figure 10 (a) gives,

$$-(M + L_1 B) + G L_2 + R_B L_2 = 0, \qquad (11)$$

from which it follows,

$$R_B = \frac{(M + L_1 B) - G L_2}{L_2}. \qquad (12)$$

Similarly, moments can be balanced about the bearing support for the decoupled model loaded with the reaction moment from
the torsional spring displayed in Figure 10(b) giving,

$$M_T - R_T L_2 = 0, \qquad (13)$$

and hence,

$$R_T = \frac{M_T}{L_2}. \qquad (14)$$

These expressions for $R_B$ and $R_T$ can now be subbed into Equations 6 and 8, respectively, resulting in solvable equations for
$\theta_{12}$ and $\theta_{22}$:

$$\theta_{12} = \frac{(M + L_1 B) - G L_2}{K L_2^2}, \qquad (15)$$


$$\theta_{22} = \frac{M_T}{K L_2^2}. \qquad (16)$$





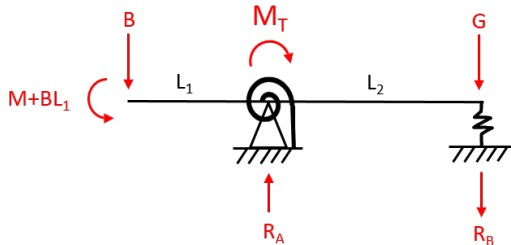

**Figure 11.** Force balance model for the whole system

Equation 10 can therefore be written in full in terms of known quantities as,

$$-\frac{M_T}{K_R} = -\frac{(M+BL_1)L_2}{3EI} - \frac{(M+BL_1)-GL_2}{KL_2^2} + \frac{M_TL_2}{3EI} + \frac{M_T}{KL_2^2},$$  (17)

and rearranged for $M_T$ as,

$$M_T = \left[\frac{(M+WL_1)L_2}{3EI} + \frac{(M+BL_1)-GL_2}{KL_2^2}\right]\left[\frac{1}{\frac{1}{K_R} + \frac{1}{KL_2^2} + \frac{L_2}{3EI}}\right].$$  (18)

The equation for the reaction moment from the torsional spring, $M_T$, has now been derived and, as such, the system is now statically determinate. A moment balance can be performed on the gearbox support over the whole system, as shown in Figure 11, to derive the reaction force at the bearing support, $R_A$,

$$R_A = \frac{M + W(L_1 + L_2) - M_T}{L_2}.$$  (19)

### 4.1 Estimating Torsional Spring Stiffness

Having derived the relevant equations for a new analytical model with moment reaction capabilities via torsional springs, it is then necessary to determine appropriate spring stiffness values in each plane. These were estimated using the FE TRB model. The body-to-ground springs representing the shaft connection to the gearbox were removed from the model and four nodes selected: one at the bedplate connection and one at the top of the bearing housing for the vertical plane and one on both sides of the bearing housing at points of mid height and mid thickness for the horizontal plane. Known moments were then applied about the horizontal and vertical axes separately and the displacement of the nodes recorded. The angle of rotation between the fixed support node and the node at the top of the housing could then be calculated and used to determine the vertical axis spring stiffness via the standard spring equation (Equation 20). Likewise, the angle of rotation about the centre of the housing between the pre and post-loaded nodal points was calculated and the torsional spring stiffness about the horizontal axis estimated. These steps are illustrated in Figures 12 and 13.

$$K = \frac{M}{\theta}$$  (20)

The two estimated spring stiffness values, approximately 208KN/rad in the horizontal and 436KN/rad in the vertical planes, were then applied to the analytical TRB model and the reaction forces at the bearing were calculated across wind profiles.





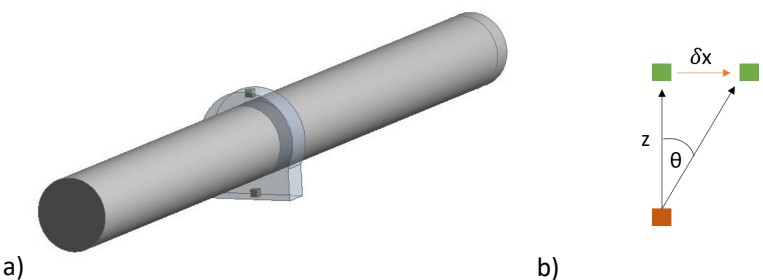

**Figure 12.** Node selection within the bearing housing for estimating torsional spring stiffness in the vertical plane.

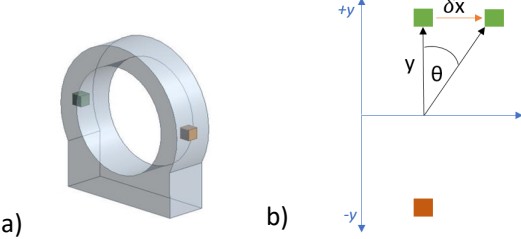

**Figure 13.** Node selection within the bearing housing for estimating torsional spring stiffness in the horizontal plane.

Examining the time series plots of the reaction forces of the FEA TRB and the analytical TRB models (presented in Figure 14)
indicates that the analytical model is capturing well the loading characteristics of the FE TRB model in both planes.

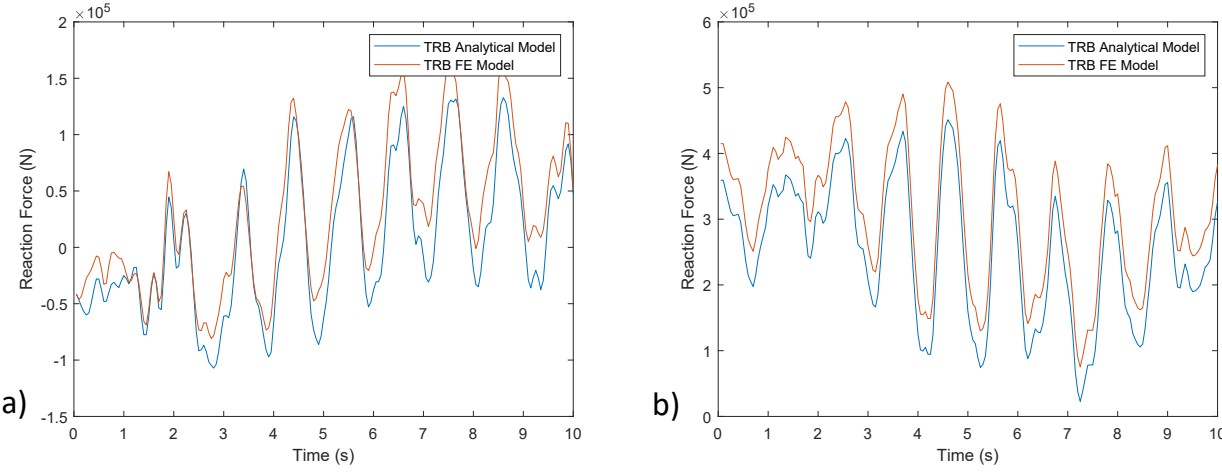

**Figure 14.** (a) Example time series of reaction force results in the horizontal plane from the analytical TRB (with torsional springs) and TRB FE models. (b) Example time series of reaction force results in the vertical plane from analytical TRB (with torsional springs) and TRB FE models.





RMSE results in this case are plotted in Figure 15. It can be seen from the plots that the inclusion of the torsional springs greatly reduces the RMSE values, as well as variance within each CPLS, between the analytical and TRB FE models in both the horizontal and vertical plane. The mean absolute error and mean correlation coefficients between resultant force magnitudes were calculated for the 2 models, mean percentage error in this case has dropped to 21.7% while the mean correlation coefficient has increased to 0.974. The results in Figure 15 show shear profile to have the strongest effect on model accuracy in both planes.

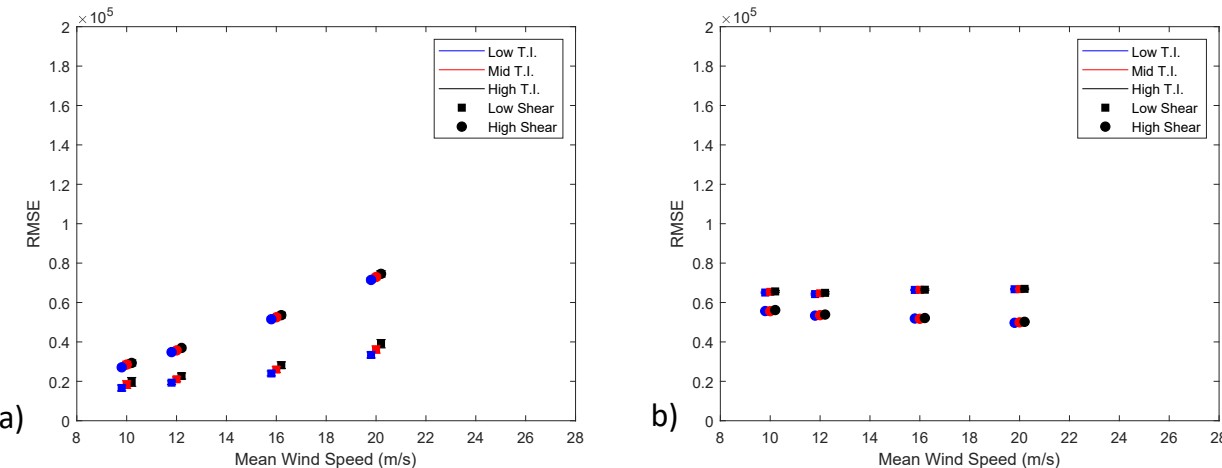

**Figure 15.** (a) RMSE between reaction forces from the analytical TRB (with torsional springs) and TRB FE model in the horizontal plane. The mean and standard deviations within each CPLS are plotted, staggered about mean wind speed for clarity. (b) RMSE reaction force results between the analytical TRB (with torsional springs) and TRB FE model in the vertical plane. The mean and standard deviations within each CPLS are plotted, staggered about mean wind speed for clarity.

## 5  Investigating mean and peak loads of SMB with both SRB and TRB

Presented results imply that the analytical SMB model of Hart et al. (2019b) and the new model developed here provide reasonable representations of SRB and TRB setups respectively. As where in previous work the mean and peak loads across operating points was considered, here these same values will be investigated for the TRB case using the analytical TRB model, with the original being referred to as the analytical SRB model.

The mean radial loading for the analytical SRB model in the previous study showed high sensitivity to shear exponent with the low shear exponent wind files resulting in larger radial loading. The low shear files saw loads between 400 and 500KN and the high shear exponent files between around 200 and 300KN. The mean loads within each CPLS remained fairly constant with small standard deviations and TI had some effect on the results with higher TI resulting in slightly higher loading.

Mean radial force and moment results for the analytical TRB model are shown in Figure 16. The presence of moment as well as force reactions can be seen to have reduced the mean radial force loading across the full envelope of wind conditions





when compared with results in Hart et al. (2019b), while also reducing the system's sensitivity to shear profile. The mean force loads within each CPLS remain fairly constant with small deviations at low mean wind speeds, although deviations increase
with increasing wind speeds.

Considering moment reactions, the majority of low shear files contribute to larger moment loading compared to high shear cases and appear to remain fairly consistent with increasing wind speeds. However, in the 20m/s case this relationship can be seen to be reversed. The variability in both shear exponent cases increases with wind speed and there are slight sensitivities to turbulence intensity.

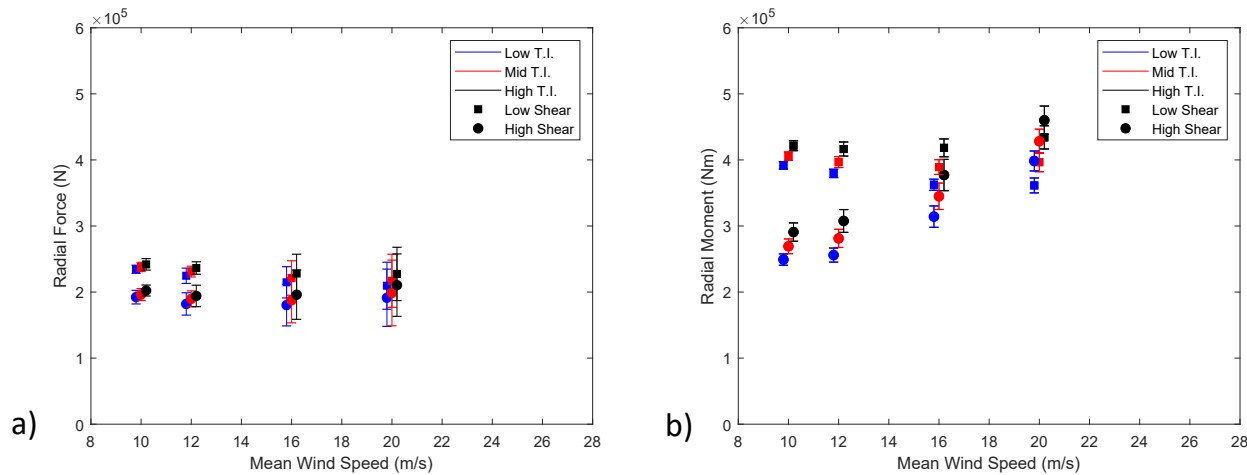

**Figure 16.** (a) Mean radial resultant force magnitudes from the analytical TRB model. Mean and standard deviations within each CPLS are plotted, staggered about mean wind speeds for clarity. (b) Mean resultant moment magnitudes from the analytical TRB model. Mean and standard deviations within each CPLS are plotted, staggered about mean wind speeds for clarity.

The analytical SRB peak radial loads presented in Hart et al. (2019b) show peak loads increasing in size and variability with increasing wind speeds. The peak loads see significant changes with TI but are most sensitive to shear exponent. All the loads fall within 500KN and 1,200KN. The mean peak radial reaction forces in the SMB system with a TRB model in fact decrease with increasing wind speed, as seen in Figure 17. The variability of peak moment loads, however, increases with wind speed. The peak radial moments show high overall sensitivity to mean wind speed and TI. The low shear exponent cases appear to
have higher peak moments with lower mean wind speeds, with high shear exponent files then having larger peak moments at higher mean wind speeds.

## 6   Conclusions

This paper has considered the validity of simplified analytical wind turbine drivetrain models for system analysis by comparing them with higher fidelity 3D FE models. The results of the comparison indicate that the existing analytical models can well

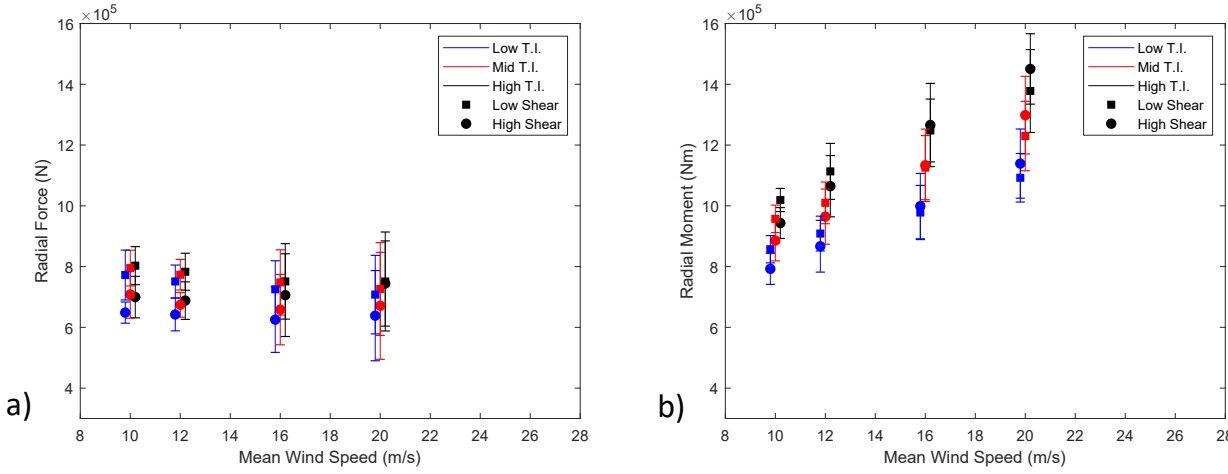

**Figure 17.** (a) Peak radial resultant force magnitudes from the analytical TRB model. Mean and standard deviations within each CPLS are plotted, staggered about mean wind speeds for clarity. (b) Peak resultant moment magnitudes from the analytical TRB model. Mean and standard deviations within each CPLS are plotted, staggered about mean wind speeds for clarity.

represent bearing reaction forces for bearings which do not react moments, while also showing it to not be suitable for cases where a bearing has moment reacting capabilities. Therefore, a second analytical model was created, through the addition of torsional springs, to represent a bearing with moment reactions. Spring stiffness's were found for this model using a static analysis of the FE model. Outputs from the new model was compared with the moment reacting 3D model, with results indicating that it offers a greatly improved tool for analysis in the moment reacting case. The developed model was then used

to consider mean and peak forces and moment reactions for this type of bearing across operating conditions. Future work on developing these models will be to considered how such reaction loading and moments properties will manifest within the bearing housing and on the rollers themselves for the two types of bearing which have been modelled thus far.

*Competing interests.*  The authors declare they have no competing interests.

*Acknowledgements.*  The authors would like to acknowledge support and advice throughout this work from Onyx Insight. This work was
funded by the EPSRC (EP/L016680/1).





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
