# Peer review of "Constructing Fast and Representative Analytical Models of Wind Turbine Main-Bearings"

_Wind Energy Science, 2020_

## Referee Comment (RC1) · Anonymous Referee #1 · 24 Jun 2020

**General**

The authors present a manuscript that deals with the calculation of main bearing re-action forces, based on previous work. They show in a very qualified way how simple approaches can also be used in the wind community. As in previous publications of the authors, the realistic wind conditions, which are used for the calculations, should be emphasized. The manuscript is well organized and written but needs major revisions in both the theoretical and practical areas.

**Specific comments**

- **The presented results are not repeatable**. Concerns arise about the used stiffness values and the practical relevance of the paper. For the FE-models,

stiffness values from ROMAX are used, but not named. The authors should give all numbers (including stiffness's, $L_1$ and $L_2$). Furthermore, the dimensions of the used bearing design are interesting for the reader. Since the main shaft will affect the FE-simulations as well, more details are needed.

- The paper compares a single main bearing system with a SRB and a TRB. It is needless to say, that different bearings need different design of the system and will have different stiffness values. The authors choose an equal design and equal values for SRB and TRB. More and detailed information are needed and a better visualization would be beneficial. The system in Figure 1 shows an axial spring, what does this spring represent?

- The simulation model needs more explanations as well. It is not clear how the shaft affects the results. The description of the manuscript is not appropriate enough to understand the results in detail.

- Implementing a torsional stiffness for the TRB seems reasonable. Nevertheless, the new approach will only deliver satisfying results, when the stiffness values from FE-simulations are given. This raises the question of the benefits of the new approach, since a simulation model is needed anyhow. Here the authors should show the benefits of the approach more clearly. It would nice to see a few examples with varying stiffness's, to see the impact.

- The authors use realistic load conditions, which makes the manuscript particularly interesting for the wind community. However, since models are compared, simple load cases, which for example only consist of a moment or a certain load, should be additionally used. This provides information about the behavior, which is not clearly explained in the current manuscript (this also increases repeatability).

**Technical corrections**

[Figure]

- In general, the introduction uses grey literature and does not show the state of the art of wind turbine main bearings. The authors should heavily improve this part of the manuscript and should focus on peer-reviewed literature instead of grey literature. Especially, the statement in line 65-68 is not supported by the grey literature (YAGI and SMALLEY) and by the previous work (HART), and should be changed appropriate.

- The Figures of the RMSE and Reaction Force are well organized. Nevertheless, in Figure 4 and 6 it is recommended to use equal values for the axis for a) and b).

---

## Referee Comment (RC2) · Anonymous Referee #2 · 1 Aug 2020

General:

The manuscript presents a method to calculate the bearing reaction forces and moments of spherical roller bearings and tapered roller bearings used as a single main bearing of wind turbines. An simplified analytical models considering the translational and rotational stiffnesses of these bearing types is shown. The benefit of this simplified approach is comprehensible. The manuscript is well structured. Major revisions are needed

Specific comments:

To be repeatable, basic parameters such as bearing dimensions and stiffnesses should be given. This is not consistently done in the manuscript

In general, more detailed illustrations of the Fe models would clearly contribute to understanding. In particular, the consideration of the contact conditions and their simplification within the FE-models should be considered in detail

The elastic behavior of the bed plate is set rigid. The author should indicate how this simplification affects the results

he physical modelling of the main bearings is not comprehensible. It seems that the spherical roller bearing has been replaced by a deformable spherical joint. It remains questionable whether this form of modelling is permissible, since the contact conditions between rolling elements and running surfaces, which varies under load, results in the characteristic non-linear stiffness of the bearing as such. In addition, no statement is made to whether the bearing clearance of the spherical roller bearing is taken into account. It is unclear how the mesh has been obtained. It is said that larger elements are used for the shaft and smaller elements are used around the bearing and bearing housing to increase accuracy at the contact regions. The mesh density is normally obtained by a convergence study. The author should indicate if this was carried out here

Also in the case of the tapered roller bearing, it is not apparent whether the contacts between the raceway surfaces and rolling elements were taken into account in the FE model. It seems as if the bearing was modelled as a piece of solid material. If this would be the case, it would have to be questioned to what extent the translational and torsional stiffness of the main bearing can be represented by the FE model. It is also indicated that the preload of the taper roller bearing is taken into account. The author should indicate how exactly the preload is considered

The analytical model is enhanced by an torsional stiffnesses of the tapered roller bearing. These stiffnesses are set constant and with that a linear stiffness behavior is indicated. In the case of roller bearings a non-linear stiffness behavior can be assumed (hertzian contact, clearance). The author should evaluate which error must be

accepted for this simplification.

The author should also add the assignment of stiffnesses K and KR in the figures

In general the results are well presented

Technical corrections:

The literature does not show the state of the art concerning modelling main bearings of wind turbines. Especially the modelling techniques used for FEM calculation should be updated.

---

## Author Comment (AC1) · 27 Aug 2020

Dear Reviewer,

Thank you for taking the time to study this paper and provide valuable constructive criticism which we believe has helped develop and strengthen this work. I have laid out all of your comments below and responded each of them in turn. Some comments touch upon the overall goal of this work and so the opening of the response has been written to clarify the main goals and overall narrative of the paper. Having received feedback on the paper we now realise that this was not outlined clearly enough in the original manuscript and so we will also be making this clearer in the updated paper.

If more information is required or we have misinterpreted any of your comments, then

please don't hesitate to get in contact and we will be happy to provide extra information to remedy this.

Best regards,

James Stirling (Lead author)

Review Comments and Author Response

Opening:

Recent work which has demonstrated important and unusual load behaviours in wind turbine main bearings has used simplified analytical representations of the drivetrain. Such simple representations will be necessary if this type of analysis is to be performed for large numbers of load cases, incorporated into fleet wide wind turbine digital twin models, used in wind farm simulation software or as part of industry standard BEM programs such as Bladed or FAST. Analytical models of these type are therefore important and already utilised in some instances. However, to date a detailed assessment of how effectively these models represent wind turbine drivetrain load reaction at the main bearing (including different bearing types) has not yet been carried out and it is therefore important to scrutinise the validity of these models and where they might apply.

As you have quite rightly pointed out, wind turbine drivetrains and main bearings in particular are specific to individual turbine designs, as such we are looking to understand in as much generality as possible how these types of analytical models may be used to represent main bearing load characteristics, without focussing on any one design case (since this would reduce the generality and applicability of results). In order to move in this direction, we have identified a need to work up through the available levels of complexity of modelling, understanding at each stage how one model represents the next in the chain. The benefit of such an approach being that at each stage, whenever a lack of agreement is found (such as in the TRB case of the present paper) small additions to the model can be sought to bring the quality of outputs back towards something which is accurate enough to be useful, while also developing knowledge about which effects can and can't be captured at each level.

In the current paper we are starting with the existing 2-dimensional, orthogonally independent, simply supported models and looking to compare with something closer to representing a real world main bearing in a wind turbine drivetrain. Since the strongest assumptions in the initial models are independence of horizontal and vertical planes (from a load perspective) and simply supported load reactions (no moment reaction, only force), we wish to compare their performance against more realistic models that don't necessarily make these assumptions. A 3-dimensional FE model avoids the orthogonality assumption. With respect to simple vs other support types we want to give the 3D model force reaction capabilities which are closer to those of real main bearings in order to assess when the simple support assumption is valid (and to consider how the simple model might be extended to compensate when it's not valid). Main bearings for wind turbines are known to have two force reaction 'types' in general. Bearings that support forces only and not moments (double row SRBs), and bearings that support both forces and moments (double row TRBs) and so simplified bearing representations are created for the 3D FE model which have these general support behaviours (without being exact models for a specific bearings).

Hence, the overarching research goal of this paper is to answer: Can analytical models be used to effectively evaluate load reaction behaviours for 3-dimensional support configurations with either moment reacting or non-moment reacting behaviours at the main support point? Tackling this question in the current paper demonstrates the validity of existing models for force reactions on the bearing as a 'unit' while also setting the stage for further work with more detailed analytical and FE model comparisons which, for example, could start evaluating internal load distributions etc.

Review comment:

The authors present a manuscript that deals with the calculation of main bearing re-action forces, based on previous work. They show in a very qualified way how simple approaches can also be used in the wind community. As in previous publications of the authors, the realistic wind conditions, which are used for the calculations, should be emphasized. The manuscript is well organized and written but needs major revisions in both the theoretical and practical areas.

Response:

We agree that a better description of the simulated wind files would help strengthen this paper. We will therefore include this in the updated manuscript and also add extra comments throughout the body of work emphasizing that the outcomes are related to realistic wind conditions and that the models remain effective over a wind turbines full operational range.

Review comment:

The presented results are not repeatable. Concerns arise about the used stiffness values and the practical relevance of the paper. For the FE-models, stiffness values from ROMAX are used, but not named. The authors should give all numbers (including stiffness's, L1 and L2). Furthermore, the dimensions of the used bearing design are interesting for the reader. Since the main shaft will affect the FE-simulations as well, more details are needed.

Response:

We agree that disclosing all dimensions and parameters of the models will help the reader gain a better understanding of the work, as well as improve reproducibility. We have spoken to industry partners and they have given the go-ahead to disclose all parameters in the paper so these will be included in the updated manuscript. A table will also be included to the paper which provides specific input forces and output results for all of the models, further helping the reader to gain an understanding of the behaviour

of the models and also to aid in repeatability.

Review comment:

The paper compares a single main bearing system with a SRB and a TRB. It is needless to say, that different bearings need different design of the system and will have different stiffness values. The authors choose an equal design and equal values for SRB and TRB. More and detailed information are needed and a better visualization would be beneficial. The system in Figure 1 shows an axial spring, what does this spring represent?

Response:

This comment is mainly addressed in the introduction of the response and centres around the goals of the study. The main purpose of the study was to compare the accuracy of the analytical models previously published by comparison with more realistic 3-dimensional models, and also test the performance when a different force reaction behaviour is present (i.e. in the case of a TRB). The models are, therefore, deliberately general and do not seek to represent any particular bearing specifically, but rather the global behaviour of different bearing types. Likewise, the rest of the drivetrain system such as the shaft and gearbox connections remain both general and similar for the two different bearing types to create a like for like study on how the bearing behaviours affect the reaction forces seen and our ability to reproduce them with simple analytical models.

We agree that the reader's understanding of the work and the FE models would be greatly improved by the inclusion of more detailed illustrations of the FE models and these will be included in the updated manuscript. We also agree that the paper would be improved through a more detailed description of the FE models and will, therefore, include information giving all dimensions, details of the mesh and how the mesh was obtained, connection types and contact conditions in the updated manuscript.

The axial spring is the stiffness equivalent of the gearbox connection in the axial direction and a description of this will be added to the paper explaining as such. This value was obtained by Onyx Insight through the use of a similar method used in this paper to obtain the equivalent spring stiffnesses in the full FE gearbox model within the Romax software.

Review comment:

The simulation model needs more explanations as well. It is not clear how the shaft affects the results. The description of the manuscript is not appropriate enough to understand the results in detail. Implementing a torsional stiffness for the TRB seems reasonable. Nevertheless, the new approach will only deliver satisfying results, when the stiffness values from FE-simulations are given. This raises the question of the benefits of the new approach, since a simulation model is needed anyhow. Here the authors should show the benefits of the approach more clearly. It would nice to see a few examples with varying stiffness's, to see the impact.

Response:

The descriptions of the FE models will be enhanced with more detail as stated above. A sensitivity analysis regarding shaft thickness is also being undertaken and included in the paper to illustrate the effect of the shaft on the results. Results of this sensitivity analysis obtained thus far indicate low sensitivity to this parameter, an important addition to the work. As stated in the introduction to the response, we are not claiming that our models directly represent a specific WT drivetrain assembly, however, all WTs have a shaft with a given stiffness and we have displayed the bearing reaction force results when this shaft stiffness is varied. Drive shafts tend to be a mostly solid piece of material, although a small hole will run throughout the shaft to allow for wiring to run through. Therefore, in our analysis we are using shaft thicknesses of 100%, 75% and 50% to conservatively cover realistic thickness (and hence stiffness) values.

The focus of this paper was not to deliver a complete and polished tool but to answer the

question of "Can analytical models accurately represent the reaction force behaviour of wind turbine main bearings?". The simple models tested and created in this body of work open the door to mass simulations and analysis in short periods of time and, thus, they could be effectively integrated into wind turbine loads simulation and monitoring at farm level during real-time operation. We agree that this could be made clearer in the paper and thus will improve the narrative in the updated manuscript.

With respect to the need for an existing FE model, during the design of WT drivetrains a detailed FE model is usually utilised. However, the company or people that do the detailed drivetrain design work, and hence have access to this FE model, will likely not share it with the wind farm operator who (for example) may be looking to develop digital twin models for their fleet. The benefit of our models is that the WF operator can request access to the non-proprietary values of equivalent stiffness values (determined using the FE model) without requiring access to the model itself. This allows for condensing of information into a form which is less commercially sensitive and allows it to be shared more widely. In addition, even where a full-blown FE model were available, it is not computationally viable to run it for each wind turbine across a wind farm where large scale studies or load/damage tracking during operation might be implemented. Furthermore, in existing certified aeroelastic codes (e.g. Bladed and FAST) structural and load analysis specifically requires for simple and fast running models of subcomponents. Models of the type developed here could therefore end up being integrated into these systems whereas FE models are simply not suitable in this context. As such we believe that there is a strong need for the models considered in this study even where an FE model (with low or high resolution) is available. You are quite right though that this discussion needs to appear in the paper in order to demonstrate the practical usefulness of its outcomes. As such this discussion will be added into the updated manuscript.

Review comment:

The authors use realistic load conditions, which makes the manuscript particularly interesting for the wind community. However, since models are compared, simple load cases, which for example only consist of a moment or a certain load, should be additionally used. This provides information about the behaviour, which is not clearly explained in the current manuscript (this also increases repeatability).

Response:

A table of inputs for a particular time step and the corresponding output results for each will be included in the updated paper to help improve the reader's understanding of the models behaviours and also improve the works reproducibility.

Review comment:

In general, the introduction uses grey literature and does not show the state of the art of wind turbine main bearings. The authors should heavily improve this part of the manuscript and should focus on peer-reviewed literature instead of grey literature. Especially, the statement in line 65-68 is not supported by the grey literature (YAGI and SMALLEY) and by the previous work (HART), and should be changed appropriate.

Response:

We also agree that more literature pertaining to wind turbine main bearings would strengthen this piece of work and this will be included in the updated manuscript. This will include [1-6], below, among others.

With respect to the second part of the comment, if there is a technical inconsistency at this stage we will be very happy to correct. However, we have struggled a little to understand the specific meaning of the comment relating to lines 65-68. It is of our understanding that the current bearing types used for main bearing in the field are most commonly double row SRBs and TRBs. We realise the bearings themselves are double rowed and we'd not added that detail before and hence have changed the sentence in the updated manuscript to include this distinction. Please feel free to respond with more details and we will endeavour to make sure our manuscript is correctly representing the

bearings used for this component. We apologise for not understanding you first time round.

Review comment:

The Figures of the RMSE and Reaction Force are well organized. Nevertheless, in Figure 4 and 6 it is recommended to use equal values for the axis for a) and b).

Response:

This has been updated as requested.

References

[1] Bosmans, J., Blockmans, B., Croes, J., Vermaut, M., and Desmet, W.: 1D-3D Nesting: Embedding reduced order flexible multibody models in system-level wind turbine drivetrain models, in: Conference for Wind Power Drives, 2019.

[2] Cardaun, M., Roscher, B., Schelenz, R., and Jacobs, G.: Analysis of Wind-Turbine Main Bearing Loads Due to Constant Yaw Misalignments 80 over a 20 Years Timespan, https://doi.org/10.3390/en12091768, www.mdpi.com/journal/energies, 2019.

[3] Gaertner, E., Rinker, J., Sethuraman, L., Zahle, F., Anderson, B., Barter, G. E., Abbas, N. J., Meng, F., Bortolotti, P., Skrzypinski, W., Scott, G. N., Feil, R., Bredmose, H., Dykes, K., Shields, M., Allen, C., and Viselli, A.: IEA Wind TCP Task 37: Definition of the IEA 15- Megawatt Offshore Reference Wind Turbine, Tech. rep., https://doi.org/10.2172/1603478, www.nrel.gov/publications.https://www.osti.gov/biblio/1603478{%}0Ahttps://www.osti.gov/servlets/purl/1603478, 2020. 85

[4] Kock, S., Jacobs, G., and Bosse, D.: Determination of Wind Turbine Main Bearing Load Distribution, in: Journal of Physics: Conference Series, vol. 1222, https://doi.org/10.1088/1742-6596/1222/1/012030, 2019.

[5] Wang, S., Nejad, A. R., and Moan, T.: On design, modelling, and

analysis of a 10-MW medium-speed drivetrain for offshore wind turbines, Wind Energy, 23, 1099–1117, https://doi.org/10.1002/we.2476, https://onlinelibrary.wiley.com/doi/abs/10.1002/we.2476, 2020b.

[6] Torsvik, J., Nejad, A. R., and Pedersen, E.: Main bearings in large offshore wind turbines: Development trends, design and analysis requirements, in: Journal of Physics: Conference Series, vol. 1037, p. 42020, https://doi.org/10.1088/1742-6596/1037/4/042020, 2018.

---

## Author Comment (AC2) · 27 Aug 2020

Dear Reviewer,

Thank you for taking the time to study this paper and provide valuable constructive criticism which we believe has helped develop and strengthen this work. I have laid out all of your comments below and responded each of them in turn (in blue). Some comments touch upon the overall goal of this work and so the opening of the response has been written to clarify the main goals and overall narrative of the paper. Having received feedback on the paper we now realise that this was not outlined clearly enough in the original manuscript and so we will also be making this clearer in the updated paper.

[Figure]

If more information is required or we have misinterpreted any of your comments, then please don't hesitate to get in contact and we will be happy to provide extra information to remedy this.

Best regards,

James Stirling (Lead author)

**Review Comments and Author Response**

**Opening:**

Recent work which has demonstrated important and unusual load behaviours in wind turbine main bearings has used simplified analytical representations of the drivetrain. Such simple representations will be necessary if this type of analysis is to be performed for large numbers of load cases, incorporated into fleet wide wind turbine digital twin models, used in wind farm simulation software or as part of industry standard BEM programs such as Bladed or FAST. Analytical models of these type are therefore important and already utilised in some instances. However, to date a detailed assessment of how effectively these models represent wind turbine drivetrain load reaction at the main bearing (including different bearing types) has not yet been carried out and it is therefore important to scrutinise the validity of these models and where they might apply.

Wind turbine drivetrains and main bearings in particular are specific to individual turbine designs, as such we are looking to understand in as much generality as possible how these types of analytical models may be used to represent main bearing load characteristics, without focussing on any one design case (since this would reduce the generality and applicability of results). In order to move in this direction, we have identified a need to work up through the available levels of complexity of modelling,

understanding at each stage how one model represents the next in the chain. The benefit of such an approach being that at each stage, whenever a lack of agreement is found (such as in the TRB case of the present paper) small additions to the model can be sought to bring the quality of outputs back towards something which is accurate enough to be useful, while also developing knowledge about which effects can and can't be captured at each level.

In the current paper we are starting with the existing 2-dimensional, orthogonally in-dependent, simply supported models and looking to compare with something closer to representing a real world main bearing in a wind turbine drivetrain. Since the strongest assumptions in the initial models are independence of horizontal and vertical planes (from a load perspective) and simply supported load reactions (no moment reaction, only force), we wish to compare their performance against more realistic models that don't necessarily make these assumptions. A 3-dimensional FE model avoids the or-thogonality assumption. With respect to simple vs other support types we want to give the 3D model force reaction capabilities which are closer to those of real main bearings in order to assess when the simple support assumption is valid (and to consider how the simple model might be extended to compensate when it's not valid). Main bearings for wind turbines are known to have two force reaction 'types' in general. Bearings that support forces only and not moments (double row SRBs), and bearings that support both forces and moments (double row TRBs) and so simplified bearing representations are created for the 3D FE model which have these general support behaviours (without being exact models for a specific bearings).

Hence, the overarching research goal of this paper is to answer: Can analytical mod-els be used to effectively evaluate load reaction behaviours for 3-dimensional support configurations with either moment reacting or non-moment reacting behaviours at the main support point? Tackling this question in the current paper demonstrates the valid-ity of existing models for force reactions on the bearing as a 'unit' while also setting the stage for further work with more detailed analytical and FE model comparisons which,

for example, could start evaluating internal load distributions etc.

**Review comment:**

Specific comments: To be repeatable, basic parameters such as bearing dimensions and stiffnesses should be given. This is not consistently done in the manuscript

**Response:**

Some parameters were initially left out of the paper due to commercial sensitivity. However, we agree that the inclusion of such parameters will help strengthen the paper by increasing repeatability and have spoken to industry partners who have given the go ahead to disclose such information in the updated manuscript. A table has also been included to the paper which provides specific input forces and output results for all of the models to help the reader to gain an understanding of the behaviour of the models and to aid in repeatability.

**Review comment:**

In general, more detailed illustrations of the FE models would clearly contribute to understanding. In particular, the consideration of the contact conditions and their simplification within the FE-models should be considered in detail

**Response:**

We agree that the reader's understanding of the work and the FE models would be greatly improved by the inclusion of more detailed illustrations of the FE models and these will be included in the updated manuscript. We also feel the paper would be improved through a more detailed description of the FE models and have, therefore, included information giving all dimensions, details of the mesh and how the mesh was obtained, connection types and contact conditions in the updated manuscript. This will not only help the reader to better understand the work undertaken but will also improve reproducibility.

[Figure]

**Review comment:**

The elastic behaviour of the bed plate is set rigid. The author should indicate how this simplification affects the results

**Response:**

We have looked at the relevant literature (e.g., [1]) concerning modelling of the bed-plate and agree that the assumptions made in this study should be brought to the reader's attention and we will therefore include a discussion of this point in the updated manuscript.

**Review comment:**

The physical modelling of the main bearings is not comprehensible. It seems that the spherical roller bearing has been replaced by a deformable spherical joint. It remains questionable whether this form of modelling is permissible, since the contact conditions between rolling elements and running surfaces, which varies under load, results in the characteristic non-linear stiffness of the bearing as such. In addition, no statement is made to whether the bearing clearance of the spherical roller bearing is taken into account. It is unclear how the mesh has been obtained. It is said that larger elements are used for the shaft and smaller elements are used around the bearing and bearing housing to increase accuracy at the contact regions. The mesh density is normally obtained by a convergence study. The author should indicate if this was carried out here

**Response:**

The purpose of this paper was to develop fast and representative models that can accurately capture the different behaviours between generic SRB and TRB load reaction behaviours when subjected to complex wind loading. As the study was designed to capture general bearing unit force reactions and not internal loads, the SRB was replaced by a deformable spherical joint. The spherical joint in ANSYS will allow the

bearing housing to deformably react forces in the X, Y and Z axes while being able to move freely in the rotational degrees of freedom. This allows the non-moment reacting behaviour of an SRB to be captured in a 3-dimensional model without going into the complexity of modelling individual rollers and hence, the global behaviour is still captured in this model in a general form. The characteristics of this simplification and the implications of it in the modelling will be discussed in the revised manuscript.

Referring back to the opening of the response, the overall goal of this study was to determine if the models in the previous study can accurately represent 3D equivalents. Although internal contact conditions between rolling elements and raceways in SRBs display non-linear stiffness behaviours, the system being modelled in this case reacts only through bedplate forces and not coupled moments and forces (where non-linear stiffness properties would determine the load 'share' between force and moment reaction contributions). As such, the overall reaction force of the bearing housing required to balance the total system remains the same regardless of internal interactions. Non-linear contact behaviour is certainly important when one is seeking to resolve distributed loads internally, but, in the current study it is the overall reaction forces which are of interest. Internal load distributions will be considered as part of the next stages of future work which will increase model complexity to that level.

Thank you for noticing we have not stated whether or not bearing clearance has been taken into account. In this instance we have assumed that there is no bearing clearance since this parameter is known to drive the internal load distribution, rather than overall reaction force. This point will be added to the updated manuscript.

We also agree that our description of how the mesh was obtained can be much improved. A convergence study was in fact carried out to determine the mesh density and a description of this will be included in the updated FE model description.

**Review comment:**

Also in the case of the tapered roller bearing, it is not apparent whether the contacts

between the raceway surfaces and rolling elements were taken into account in the FE model. It seems as if the bearing was modelled as a piece of solid material. If this would be the case, it would have to be questioned to what extent the translational and torsional stiffness of the main bearing can be represented by the FE model. It is also indicated that the preload of the taper roller bearing is taken into account. The author should indicate how exactly the preload is considered.

The analytical model is enhanced by an torsional stiffnesses of the tapered roller bearing. These stiffnesses are set constant and with that a linear stiffness behavior is indicated. In the case of roller bearings a non-linear stiffness behavior can be assumed (hertzian contact, clearance). The author should evaluate which error must be accepted for this simplification.

**Response:**

Both of these comments tie into the opening of this response and the main goals of this study. You are correct that the load shared between force and moment reactions within the TRB will be determined by the stiffness behaviour (as was touched on above) in the bearing, however, TRB are known to have only weak non-linear behaviour (with a deflection exponent value of 1.1) and TRBs, along with CRBs, are often approximated as linear in their load response. This type of bearing can therefore be approximated to behave like linear steel sections in the FE model and then, since it is the type of load reaction (forces and moments) rather than any one specific design, we have approximated this with a piece of solid material. This is in-line with the stated goal of the paper outlined in the first part of this response (and to be added very clearly into the revised manuscript) to explore how well analytical models might recreate the loads experienced by a support which reacts both forces and moments. This discussion of the modelling assumptions employed, and their viability should have been included in the original manuscript and so we are very grateful you have brought this oversight to our attention. To be clear, we are not proposing that the FE models we employ here should be used to represent real world TRBs, we are developing a methodology from which

someone can use an accurate FE representation of their TRB bearing to develop fast and representative analytical models suitable for use in large numbers of load analysis cases, development of digital twin models across a large turbine fleet or similar applications where computationally expensive FE analysis is not viable. The results of this work demonstrate that, up to the level of models employed here, this can be done for both SRB and TRB reaction behaviour types. With respect to the added torsional springs being linear, under small deformations (such as those present in bearings) a torsional spring is equivalent to a pair of parallel linear springs and hence the fact that TRB contact behaviour is only very weakly non-linear indicates that a linear torsional spring is a reasonable approximation. This point will be revisited in future work where internal forces and deformations are considered as modelling complexity is increased. We will also ensure that the above points are clear in the updated manuscript.

As we are interested in the overall forces and moments, the bearing preload effectively gives further justification for assuming the bearing and housing are a solid piece of material (no clearance) – we'll make this point clearer in the updated manuscript.

**Review comment:**

The author should also add the assignment of stiffnesses K and KR in the figures

**Response:**

K1, K2 and KR will be added to the figures and the values given in the figure description.

**Review comment:**

In general the results are well presented

The literature does not show the state of the art concerning modelling main bearings of wind turbines. Especially the modelling techniques used for FEM calculation should be updated.

**Response:**

We also agree that more literature pertaining to the modelling of wind turbine main bearings would strengthen this piece of work and this will be included in the updated manuscript. This will include ref [2-7] among others.

**References**

[1] Wang, S., Nejad, A. R., Bachynski, E. E., and Moan, T.: Effects of bedplate flexibility on drivetrain dynamics: Case study of a 10MW spar 95 type floating wind turbine, Renewable Energy, 161, 808–824, https://doi.org/10.1016/j.renene.2020.07.148, https://linkinghub.elsevier. com/retrieve/pii/S0960148120312246, 2020a.

[2] Bosmans, J., Blockmans, B., Croes, J., Vermaut, M., and Desmet, W.: 1D-3D Nesting: Embedding reduced order flexible multibody models in system-level wind turbine drivetrain models, in: Conference for Wind Power Drives, 2019.

[3] Cardaun, M., Roscher, B., Schelenz, R., and Jacobs, G.: Analysis of Wind-Turbine Main Bearing Loads Due to Constant Yaw Misalignments 80 over a 20 Years Timespan, https://doi.org/10.3390/en12091768, www.mdpi.com/journal/energies, 2019.

[4] Gaertner, E., Rinker, J., Sethuraman, L., Zahle, F., Anderson, B., Barter, G. E., Abbas, N. J., Meng, F., Bortolotti, P., Skrzypinski, W., Scott, G. N., Feil, R., Bredmose, H., Dykes, K., Shields, M., Allen, C., and Viselli, A.: IEA Wind TCP Task 37: Definition of the IEA 15- Megawatt Offshore Reference Wind Turbine, Tech. rep., https://doi.org/10.2172/1603478, www.nrel.gov/publications.https://www.osti.gov/biblio/1603478

[5] Kock, S., Jacobs, G., and Bosse, D.: Determination of Wind Turbine Main Bearing Load Distribution, in: Journal of Physics: Conference Series, vol. 1222, https://doi.org/10.1088/1742-6596/1222/1/012030, 2019.

[Figure]

[6] Wang, S., Nejad, A. R., and Moan, T.: On design, modelling, and analysis of a 10-MW medium-speed drivetrain for offshore wind turbines, Wind Energy, 23, 1099–1117, https://doi.org/10.1002/we.2476, https://onlinelibrary.wiley.com/doi/abs/10.1002/we.2476, 2020b. [7] Torsvik, J., Nejad, A. R., and Pedersen, E.: Main bearings in large offshore wind turbines: Development trends, design and analysis requirements, in: Journal of Physics: Conference Series, vol. 1037, p. 42020, https://doi.org/10.1088/1742-6596/1037/4/042020, 2018.

---

## Author Comment (AC3) · 27 Aug 2020

Dear Charlotte,

Thank you for handling this submission and supporting the review process for this work.

It was a pleasure to see that both reviewers had studied the paper in detail and given valuable feedback and suggestions. Their comments have allowed us to understand where the manuscript needed tightening up and the narrative and goals made more clear. In addition, there were some parts of the modelling and analysis that needed more detail to ensure clarity and reproducibility.

Work is currently underway to address all the comments by both reviewers and detailed responses to each has been included alongside our plan to implement the suggested

changes in the 'Response to reviewer' documents which we have also uploaded.

Main updates which are being undertaken are: improved clarity regarding the goals and narrative of this paper, expanded introduction to include a broader discussion of related work, disclosure of dimensions and parameters used in models, improved descriptions of the FE models and mesh sizing procedure, new shaft thickness sensitivity analysis and the addition of a table of numerical model outputs for given inputs to aid reproducibility.

If any further information is required regarding the manuscript discussion and responses, we will be most happy to engage further.

Best regards,

James Stirling

---

## Author Response (AR1)

**Authors' Response: Constructing Fast and Accurate Analytical Models for Wind Turbine Main-Bearings**

James Stirling1, Edward Hart2, and Abbas Kazemi Amiri2

1Wind and Marine Energy Systems CDT, EEE, University of Strathclyde, Glasgow, UK 2Wind Energy and Control Centre, EEE, University of Strathclyde, Glasgow, UK

Correspondence: James Stirling (j.stirling@strath.ac.uk)

Dear reviewers, we would like to thank you again for taking the time to review our manuscript. Your comments made us realise that important clarifications and additional details were required and so we have re-written a great deal of the paper to ensure much greater clarity and more detail is included with respect to the aims, rationales and models used in this work. Furthermore,

5 we discovered that reference frame related issues had been causing originally reported results to be worse than they actually are. Hence, in the updated manuscript you will see that the new model now brings errors related to the moment-reacting support down from over 20% (without torsional springs) to less than 2% (after addition of torsional springs). All equations and code have been thoroughly scrutinised to ensure no issues remain.

Much of the following is taken from our original responses posted to the interactive discussion on the WES page for this paper. Here those same discussions and details are included, with specific additional information about the changes which have now been made, included in red.

Section 1 contains the response preamble which featured in the original responses to both reviewers. The following two sections then address the comments of the two reviewers respectively and take the following form:

- Reviewer's comment

- 15 Authors' response from interactive discussion
  - Authors' changes to manuscript

The marked-up manuscript is appended onto the end of this document and all line number references refer to this version of the updated manuscript.

**1 Authors' Response Preamble**

20 Recent work which has demonstrated important and unusual load behaviours in wind turbine main bearings has used simplified analytical representations of the drivetrain. Such simple representations will be necessary if this type of analysis is to be performed for large numbers of load cases, incorporated into fleet wide wind turbine digital twin models, used in wind farm simulation software or as part of industry standard BEM programs such as Bladed or FAST. Analytical models of these type are therefore important and already utilised in some instances. However, to date a detailed assessment of how effectively these

25 models represent wind turbine drivetrain load reaction at the main bearing (including different bearing types) has not yet been carried out and it is therefore important to scrutinise the validity of these models and where they might apply.

Wind turbine drivetrains and main bearings in particular are specific to individual turbine designs, as such we are looking to understand in as much generality as possible how these types of analytical models may be used to represent main bearing load characteristics, without focussing on any one design case (since this would reduce the generality and applicability of results).

- 30 In order to move in this direction, we have identified a need to work up through the available levels of complexity of modelling, understanding at each stage how one model represents the next in the chain. The benefit of such an approach being that at each stage, whenever a lack of agreement is found (such as in the TRB case of the present paper) small additions to the model can be sought to bring the quality of outputs back towards something which is accurate enough to be useful, while also developing knowledge about which effects can and can't be captured at each level.
- 35 In the current paper we are starting with the existing 2-dimensional, orthogonally independent, simply supported models and looking to compare with something closer to representing a real world main bearing in a wind turbine drivetrain. Since the strongest assumptions in the initial models are independence of horizontal and vertical planes (from a load perspective) and simply supported load reactions (no moment reaction, only force), we wish to compare their performance against more realistic models that don't necessarily make these assumptions. A 3-dimensional FE model avoids the orthogonality assumption. With
- 40 respect to simple vs other support types we want to give the 3D model force reaction capabilities which are closer to those of real main bearings in order to assess when the simple support assumption is valid (and to consider how the simple model might be extended to compensate when it's not valid). Main bearings for wind turbines are known to have two force reaction 'types' in general. Bearings that support forces only and not moments (double row SRBs), and bearings that support both forces and moments (double row TRBs) and so simplified bearing representations are created for the 3D FE model which have these general support behaviours (without being exact models for a specific bearings).

Hence, the overarching research goal of this paper is to answer: Can analytical models be used to effectively evaluate load reaction behaviours for 3-dimensional support configurations with either moment reacting or non-moment reacting behaviours at the main support point? Tackling this question in the current paper demonstrates the validity of existing models for force reactions on the bearing as a 'unit' while also setting the stage for further work with more detailed analytical and FE model comparisons which, for example, could start evaluating internal load distributions etc.

- The text of the paper has been extensively revised to ensure the points made in this discussion are very clear throughout the manuscript. In particular, this includes important improvements to the Introduction, Background and FE Model sections, as well as the inclusion of a new Discussion section at the end of the manuscript.

**2 Authors' Response to Reviewer 1**

50

55 1. The authors present a manuscript that deals with the calculation of main bearing re- action forces, based on previous work. They show in a very qualified way how simple approaches can also be used in the wind community. As in previous publications of the authors, the realistic wind conditions, which are used for the calculations, should be emphasized. The manuscript is well organized and written but needs major revisions in both the theoretical and practical areas.

We agree that a better description of the simulated wind files would help strengthen this paper. We will therefore include
this in the updated manuscript and also add extra comments throughout the body of work emphasizing that the outcomes are related to realistic wind conditions and that the models remain effective over a wind turbines full operational range.

- The description of the wind fields used in the study has been updated and expanded and can be found in lines 98-111.

**2.** The presented results are not repeatable. Concerns arise about the used stiffness values and the practical relevance of the paper. For the FE-models, stiffness values from ROMAX are used, but not named. The authors should give all numbers (including stiffness's, L1 and L2). Furthermore, the dimensions of the used bearing design are interesting for the reader. Since

the main shaft will affect the FE-simulations as well, more details are needed.

65

-We agree that disclosing all dimensions and parameters of the models will help the reader gain a better understanding of the work, as well as improve reproducibility. We have spoken to industry partners and they have given the go-ahead to disclose all parameters in the paper so these will be included in the updated manuscript. A table will also be included to the paper which

70 provides specific input forces and output results for all of the models, further helping the reader to gain an understanding of the behaviour of the models and also to aid in repeatability.

-All model parameters have been included in Table 1. A table has also been included in Appendix A containing hub loading inputs and corresponding model outputs at various time steps.

3. The paper compares a single main bearing system with a SRB and a TRB. It is needless to say, that different bearings need different design of the system and will have different stiffness values. The authors choose an equal design and equal values for SRB and TRB. More and detailed information are needed and a better visualization would be beneficial. The system in Figure 1 shows an axial spring, what does this spring represent?

This comment is mainly addressed in the introduction of the response and centres around the goals of the study. The main purpose of the study was to compare the accuracy of the analytical models previously published by comparison with more
realistic 3-dimensional models, and also test the performance when a different force reaction behaviour is present (i.e. in the case of a TRB). The models are, therefore, deliberately general and do not seek to represent any particular bearing specifically, but rather the global behaviour of different bearing types. Likewise, the rest of the drivetrain system such as the shaft and gearbox connections remain both general and similar for the two different bearing types to create a like for like study on how the bearing behaviours affect the reaction forces seen and our ability to reproduce them with simple analytical models. We agree

85 that the reader's understanding of the work and the FE models would be greatly improved by the inclusion of more detailed illustrations of the FE models and these will be included in the updated manuscript. We also agree that the paper would be improved through a more detailed description of the FE models and will, therefore, include information giving all dimensions, details of the mesh and how the mesh was obtained, connection types and contact conditions in the updated manuscript. The axial spring is the stiffness equivalent of the gearbox connection in the axial direction and a description of this will be added to

90 the paper explaining as such. This value was obtained by Onyx Insight through the use of a similar method used in this paper to obtain the equivalent spring stiffnesses in the full FE gearbox model within the Romax software.

- Descriptions of the axial spring are provided in lines 127-128 and 168-172 and the spring stiffness value is included in Table 1. A more detailed description of the FE models has been included in Section 3 and the reasons for their similarities explained in lines 142-151. Figures 2 and 4 have also been added to provide full images of the FE models.

- 4. The simulation model needs more explanations as well. It is not clear how the shaft affects the results. The description of the manuscript is not appropriate enough to understand the results in detail. Implementing a torsional stiffness for the TRB seems reasonable. Nevertheless, the new approach will only deliver satisfying results, when the stiffness values from FE-simulations are given. This raises the question of the benefits of the new approach, since a simulation model is needed anyhow. Here the authors should show the benefits of the approach more clearly. It would nice to see a few examples with varying stiffness's, to see the impact.
  - The descriptions of the FE models will be enhanced with more detail as stated above. A sensitivity analysis regarding shaft thickness is also being undertaken and included in the paper to illustrate the effect of the shaft on the results. Results of this sensitivity analysis obtained thus far indicate low sensitivity to this parameter, an important addition to the work. As stated in the introduction to the response, we are not claiming that our models directly represent a specific WT drivetrain assembly,
- 105 however, all WTs have a shaft with a given stiffness and we have displayed the bearing reaction force results when this shaft stiffness is varied. Drive shafts tend to be a mostly solid piece of material, although a small hole will run throughout the shaft to allow for wiring to run through. Therefore, in our analysis we are using shaft thicknesses of 100%, 75% and 50% to conservatively cover realistic thickness (and hence stiffness) values.
  - The focus of this paper was not to deliver a complete and polished tool but to answer the question of "Can analytical models accurately represent the reaction force behaviour of wind turbine main bearings?". The simple models tested and created in this body of work open the door to mass simulations and analysis in short periods of time and, thus, they could be effectively integrated into wind turbine loads simulation and monitoring at farm level during real-time operation. We agree that this could be made clearer in the paper and thus will improve the narrative in the updated manuscript. With respect to the need for an existing FE model, during the design of WT drivetrains a detailed FE model is usually utilised. However, the company or
  - 115 people that do the detailed drivetrain design work, and hence have access to this FE model, will likely not share it with the wind farm operator who (for example) may be looking to develop digital twin models for their fleet. The benefit of our models is that the WF operator can request access to the non-proprietary values of equivalent stiffness values (determined using the FE model) without requiring access to the model itself. This allows for condensing of information into a form which is less commercially sensitive and allows it to be shared more widely. In addition, even where a full-blown FE model were available,
  - 120 it is not computationally viable to run it for each wind turbine across a wind farm where large scale studies or load/damage tracking during operation might be implemented. Furthermore, in existing certified aeroelastic codes (e.g. Bladed and FAST) structural and load analysis specifically requires for simple and fast running models of subcomponents. Models of the type

developed here could therefore end up being integrated into these systems whereas FE models are simply not suitable in this context. As such we believe that there is a strong need for the models considered in this study even where an FE model (with

low or high resolution) is available. You are quite right though that this discussion needs to appear in the paper in order to 125 demonstrate the practical usefulness of its outcomes. As such this discussion will be added into the updated manuscript.

- A more detailed description of the FE models has been included in Section 3. A sensitivity analysis on shaft thickness was carried out to determine how shaft stiffness affects the simulation results. The sensitivity analysis results are included in Appendix A. Practicalities of this approach (given you need some access to an FE model) are treated in the new 'Discussion' section at the end of the paper.

130

5. The authors use realistic load conditions, which makes the manuscript particularly interesting for the wind community. However, since models are compared, simple load cases, which for example only consist of a moment or a certain load, should be additionally used. This provides information about the behaviour, which is not clearly explained in the current manuscript (this also increases repeatability).

- A table of inputs for a particular time step and the corresponding output results for each will be included in the updated paper 135 to help improve the reader's understanding of the models behaviours and also improve the works reproducibility.

- Table A1 has been added to Appendix A which provides a variety of hub loading inputs (in orthogonal force and moment components) and the reaction force results for each model.

6. In general, the introduction uses grey literature and does not show the state of the art of wind turbine main bearings. The authors should heavily improve this part of the manuscript and should focus on peer-reviewed literature instead of grey 140 literature. Especially, the statement in line 65-68 is not supported by the grey literature (YAGI and SMALLEY) and by the previous work (HART), and should be changed appropriate.

- We also agree that more literature pertaining to wind turbine main bearings would strengthen this piece of work and this will be included in the updated manuscript. This will include [1-6], below, among others.

- 145 With respect to the second part of the comment, if there is a technical inconsistency at this stage we will be very happy to correct. However, we have struggled a little to understand the specific meaning of the comment relating to lines 65-68. It is of our understanding that the current bearing types used for main bearing in the field are most commonly double row SRBs and TRBs. We realise the bearings themselves are double rowed and we'd not added that detail before and hence have changed the sentence in the updated manuscript to include this distinction. Please feel free to respond with more details and we will
- 150

endeavour to make sure our manuscript is correctly representing the bearings used for this component. We apologise for not understanding you first time round.

- Section 2 has been expanded to include summaries of recent modelling pertaining to wind turbine main bearings in the literature (lines 64-85). The introduction has also been expanded to include a better description of the study and better define the overarching goal. The statement which was previously in line 65-68 has been altered and can now be found in lines 135-140.

155 We hope this clears up the issue here but if not then we'll be very happy to take any further comments into account.

**7.** The Figures of the RMSE and Reaction Force are well organized. Nevertheless, in Figure 4 and 6 it is recommended to use equal values for the axis for a) and b).

- This has been updated as requested.

- Figures 4 and 6 have been updated accordingly.

**160 3 Authors' Response to Reviewer 2**

**1.** Specific comments: To be repeatable, basic parameters such as bearing dimensions and stiffnesses should be given. This is not consistently done in the manuscript

- Some parameters were initially left out of the paper due to commercial sensitivity. However, we agree that the inclusion of such parameters will help strengthen the paper by increasing repeatability and have spoken to industry partners who have given

165 the go ahead to disclose such information in the updated manuscript. A table has also been included to the paper which provides specific input forces and output results for all of the models to help the reader to gain an understanding of the behaviour of the models and to aid in repeatability.

- All model parameters have been included in Table 1. A table has also been included in Appendix A containing hub loading inputs and corresponding model outputs at various time steps to further aid reproducibility.

**2.** In general, more detailed illustrations of the FE models would clearly contribute to understanding. In particular, the consideration of the contact conditions and their simplification within the FE-models should be considered in detail

- We agree that the reader's understanding of the work and the FE models would be greatly improved by the inclusion of more detailed illustrations of the FE models and these will be included in the updated manuscript. We also feel the paper would be improved through a more detailed description of the FE models and have, therefore, included information giving

175

all dimensions, details of the mesh and how the mesh was obtained, connection types and contact conditions in the updated manuscript. This will not only help the reader to better understand the work undertaken but will also improve reproducibility.

- More detailed descriptions of the FE models have been included in Section 3. Figures 2 and 4 have also been added to provide full images of the FE models.

3. The elastic behaviour of the bed plate is set rigid. The author should indicate how this simplification affects the results

- 180 We have looked at the relevant literature (e.g., [1]) concerning modelling of the bedplate and agree that the assumptions made in this study should be brought to the reader's attention and we will therefore include a discussion of this point in the updated manuscript.
  - The effects of assuming a rigid bedplate have been included in lines 166-167, citing relevant literature.

4. The physical modelling of the main bearings is not comprehensible. It seems that the spherical roller bearing has been replaced by a deformable spherical joint. It remains questionable whether this form of modelling is permissible, since the contact conditions between rolling elements and running surfaces, which varies under load, results in the characteristic non-linear stiffness of the bearing as such. In addition, no statement is made to whether the bearing clearance of the spherical roller bearing is taken into account. It is unclear how the mesh has been obtained. It is said that larger elements are used for the shaft and smaller elements are used around the bearing and bearing housing to increase accuracy at the contact regions. The mesh density is normally obtained by a convergence study. The author should indicate if this was carried out here

- The purpose of this paper was to develop fast and representative models that can accurately capture the different behaviours between generic SRB and TRB load reaction behaviours when subjected to complex wind loading. As the study was designed to capture general bearing unit force reactions and not internal loads, the SRB was replaced by a deformable spherical joint. The spherical joint in ANSYS will allow the bearing housing to deformably react forces in the X, Y and Z axes while being able to move freely in the rotational degrees of freedom. This allows the non-moment reacting behaviour of an SRB to be captured in a 2 dimensional model without going into the complexity of modelling individual rollers and hence, the global

- captured in a 3-dimensional model without going into the complexity of modelling individual rollers and hence, the global behaviour is still captured in this model in a general form. The characteristics of this simplification and the implications of it in the modelling will be discussed in the revised manuscript.
- Referring back to the opening of the response, the overall goal of this study was to determine if the models in the previous study can accurately represent 3D equivalents. Although internal contact conditions between rolling elements and raceways in SRBs display non-linear stiffness behaviours, the system being modelled in this case reacts only through bedplate forces and not coupled moments and forces (where nonlinear stiffness properties would determine the load 'share' between force and moment reaction contributions). As such, the overall reaction force of the bearing housing required to balance the total system remains the same regardless of internal interactions. Non-linear contact behaviour is certainly important when one is seeking to
- 205

195

5 resolve distributed loads internally, but, in the current study it is the overall reaction forces which are of interest. Internal load distributions will be considered as part of the next stages of future work which will increase model complexity to that level.

Thank you for noticing we have not stated whether or not bearing clearance has been taken into account. In this instance we have assumed that there is no bearing clearance since this parameter is known to drive the internal load distribution, rather than overall reaction force. This point will be added to the updated manuscript.

210 We also agree that our description of how the mesh was obtained can be much improved. A convergence study was in fact carried out to determine the mesh density and a description of this will be included in the updated FE model description.

7

- The descriptions of the FE models have been improved and include details on bearing clearance assumptions for both the DSRB and DTRB models and Figure 2 and 4 have been added which display the models in their entirety. A description of how the mesh was obtained is also included in lines 176-178 and 191-194. A paragraph considering bearing contact assumptions with respect to both models has also been included (lines 195-207).

215

**5.** Also in the case of the tapered roller bearing, it is not apparent whether the contacts between the raceway surfaces and rolling elements were taken into account in the FE model. It seems as if the bearing was modelled as a piece of solid material. If this would be the case, it would have to be questioned to what extent the translational and torsional stiffness of the main bearing can be represented by the FE model. It is also indicated that the preload of the taper roller bearing is taken into account. The author should indicate how exactly the preload is considered.

220

The analytical model is enhanced by an torsional stiffnesses of the tapered roller bearing. These stiffnesses are set constant and with that a linear stiffness behavior is indicated. In the case of roller bearings a non-linear stiffness behavior can be assumed (hertzian contact, clearance). The author should evaluate which error must be accepted for this simplification.

- Both of these comments tie into the opening of this response and the main goals of this study. You are correct that the load 225 shared between force and moment reactions within the TRB will be determined by the stiffness behaviour (as was touched on above) in the bearing, however, TRB are known to have only weak non-linear behaviour (with a deflection exponent value of 1.1) and TRBs, along with CRBs, are often approximated as linear in their load response. This type of bearing can therefore be approximated to behave like linear steel sections in the FE model and then, since it is the type of load reaction (forces and moments) rather than any one specific design, we have approximated this with a piece of solid material. This is in-line 230 with the stated goal of the paper outlined in the first part of this response (and to be added very clearly into the revised manuscript) to explore how well analytical models might recreate the loads experienced by a support which reacts both forces and moments. This discussion of the modelling assumptions employed, and their viability should have been included in the original manuscript and so we are very grateful you have brought this oversight to our attention. To be clear, we are not proposing that the FE models we employ here should be used to represent real world TRBs, we are developing a methodology 235 from which someone can use an accurate FE representation of their TRB bearing to develop fast and representative analytical models suitable for use in large numbers of load analysis cases, development of digital twin models across a large turbine

- fleet or similar applications where computationally expensive FE analysis is not viable. The results of this work demonstrate that, up to the level of models employed here, this can be done for both SRB and TRB reaction behaviour types. With respect to the added torsional springs being linear, under small deformations (such as those present in bearings) a torsional spring is
  equivalent to a pair of parallel linear springs and hence the fact that TRB contact behaviour is only very weakly non-linear
- indicates that a linear torsional spring is a reasonable approximation. This point will be revisited in future work where internal forces and deformations are considered as modelling complexity is increased. We will also ensure that the above points are clear in the updated manuscript.

As we are interested in the overall forces and moments, the bearing preload effectively gives further justification for assuming the bearing and housing are a solid piece of material (no clearance) – we'll make this point clearer in the updated manuscript. - As in the previous response, the descriptions of the FE models have been improved and now include details on bearing clearance assumptions etc. The mentioned additional paragraph discussing bearing contact assumptions also helps clarify some of the points raised here (lines 195-207). The comment relating to pre-loading had been made clearer, it is essentially another justification for zero clearance being a reasonable assumption in the DTRB case (line 183).

- 250 6. The author should also add the assignment of stiffnesses K and KR in the figures
  - K1, K2 and KR will be added to the figures and the values given in the figure description.
  - K1, K2 and KR have been added to all relevant figures and their values given in Table 1 and line 281.

**Constructing Fast and Representative Analytical Models of Wind Turbine Main-Bearings**

James Stirling1, Edward Hart2, and Abbas Kazemi Amiri2

1Wind and Marine Energy Systems CDT, EEE, University of Strathclyde, Glasgow, UK 2Wind Energy and Control Centre, EEE, University of Strathclyde, Glasgow, UK

**Correspondence:** James Stirling, Wind and Marine Energy Systems CDT, EEE, University of Strathclyde, Glasgow, UK (j.stirling@strath.ac.uk)

**Abstract.** This paper considers the modelling of wind turbine main bearings main-bearings using analytical models. The validity of simplified analytical representations used in existing work is explored by comparing main bearing main-bearing force reactions with those obtained from higher fidelity 3D finite element models. Results indicate that there is good agreement between the analytical and 3D models in the case of a non-moment-reacting case non-moment reacting support (such as for

- 5 a spherical roller bearingdouble row spherical roller bearings), but, the same does not hold in the moment reacting case (such as for double row tapered roller bearings). Therefore, a new analytical model is developed in which moment reactions at the main bearing main-bearing are captured through the addition of torsional springs. This latter model is shown to significantly improve the agreement between analytical and 3D models in the moment reacting case. The new analytical model is then used to investigate load characteristics, in terms of forces and moments, for this type of main bearing main-bearing across different
- 10 operating points and wind conditions.

**1 Introduction**

Wind energy provides an important and growing contribution to the European energy market, with 205GW installed as of 2019 - accounting for 15% of consumed electricity (Wind Europe, 2020). As part of this growth, more wind farms are being planned and constructed offshore to take advantage of higher wind speeds and more available construction space (Junginger et al.,

15 2004). Recent trends show dramatic falls in the cost of offshore wind, as been mirrored in the UK's contract for difference auctions which have seen prices drop to £57.50/MWh (UK Government, 2017) and even lower.

With turbines moving further offshore and a need to bring costs down<del>means that</del>, reducing operation and maintenance costs, which can be as high as 35% of the total lifetime costs of a project, is becoming increasingly important for wind farm operators (Sinha and Steel, 2015). This in turn effects technology design and selection and puts pressure on original equipment man-

20 ufacturers (OEMs) and operators to improve turbine reliability. As such, reliability and failure rate considerations have received much attention in the literature (Walford, 2006; Tavner et al., 2007; Wilkinson, 2011)(Tavner et al., 2007; Wilkinson, 2011; Artigao et al.,

One turbine component with relatively high failures failure rates and associated downtime is the main bearing main-bearing (MB). MBs are becoming recognised as an important component for which failures need to be better understood and reliability

- 25 improved (Keller et al., 2016; ?)(Keller et al., 2016; Hart et al., 2020). MB failure rates have been reported as being up to as high as 30% (Hart et al., 2019) across a 20 year lifetime, with some wind farms having reported MBs failing in less than 6 years (Sethuraman et al., 2015). Recent work which has demonstrated important and unusual load behaviours in wind turbine MBs (Hart et al., 2019; Hart, 2020) implements simplified analytical representations of the drivetrain. Such representations are necessary if this type of analysis is to be performed across large numbers of load cases, incorporated into fleet wide modelling,
- 30 or into industry standard simulation software (e.g. Bladed and Fast). These types of analytical models are therefore important and already being utilised and, as such, a detailed assessment of how effectively they represent wind turbine drivetrain load response at the MB for different bearing types is an important next step in their development.

Preventing premature failures of main-bearings would therefore be an important contribution to reducing operating costs of wind farms. As part of analyses which try to understand the loading conditions of MBs in wind turbines (in order to better understand their operational conditions and load characteristsics), detailed model-based investigations are required. Work of this type exists in the literature (Hart et al., 2019) in which analytical models are used to consider MB loading.

35

55

Wind turbine drivetrains and MBs in particular are specific to individual turbine designs. As such, it is beneficial to understand in as much generality as possible how existing simple representations may be used to study MB load response, without focusing on any design case (since this would reduce the generality of results). In order to move in this direction, it is

40 necessary to work through levels of modelling complexity, understanding at each stage how well a given model represents the next in the chain. This approach also develops knowledge about which effects can be adequately captured at a given level of model complexity, helping inform decisions with respect to model selection for specific applications.

This paper considers the validity of simplified analytical drivetrain representations of the type used in these load studies by comparisons with higher fidelity an important step in the overall modelling chain; starting with existing 2D, orthogonally

- 45 independent, simply supported models and looking to compare with higher fidelity models which are closer to representing real-world wind turbine MBs. The strongest assumptions in the existing models are: independence of horizontal and vertical planes (from a load perspective) and simply supported load reactions (i.e. the bearing does not support moment loads). Therefore, this work seeks to compare their performance with more realistic models that remove one or both of these assumptions. More explicitly, 3D finite element models finite-element (FE) modelling removes the 2D and orthogonality assumptions. With
- 50 respect to simple versus other supports, MBs for wind turbines have two 'types' of reaction behaviour in general; those that support forces only and not moments (e.g. double row spherical roller bearings (DSRBs)), and those that support both forces and moments (e.g. double row tapered roller bearings (DTRBs)). 3D FE models will therefore be considered which have reaction behaviours that emulate each of the two types. Hence, the overarching goal of this paper is to explore the question:

Can analytical models be used to effectively evaluate load reactions for 3-dimensional main-bearing support configurations with either moment reacting or non-moment reacting behaviours?

Section 2 provides a description of the summarises previous work undertaken in this area. Section 3 then introduces the higher fidelity 3D models which will be used to validate the analytical models before presenting compare with analytical model outputs. Section 4 presents the results of the comparison. In section 4, with Section 5 then extending the analytical model

is adapted to include moment reactions at the MB, before comparing the models again. Section 5 applies . In Section 6 the

60 new analytical model is used to study load behaviours for this bearing type. Finally, Section 6-7 discusses some practicalities surrounding the application of these models before Section 8 presents the conclusions of this work.

**2 Previous WorkBackground**

A proper understanding of Despite having received less attention than other drivetrain components, there have been a number of high quality research papers which include modelling and analysis of wind turbine MBs. Cardaun et al. (2019) use a multibody simulation model with flexible components in SIMPACK to investigate main-bearing loading requires full consideration of the complex load environment with which the bearing is interacting. This work expands from work completed previously (Hart et al., 2019) in which hub loading time histories were generated using multi-body aero-elastic software and injected into simplified 2 dimensional models of realistic MB set-ups to determine MB operational loading. loads for a yawed turbine.

- 70 It was found that yawed inflow has an asymmetric effect on main-bearing loading and fatigue, with the possibility of either increasing or decreasing loading and load fluctuations depending on yaw direction relative to inflow. Bosmans et al. (2019) represent the drivetrain system as lumped parameter components in order to keep degrees of freedom low and increase the speed of simulations, bearings are modelled as linear springs. The study showed differences between port-based and 1D-3D nesting models. In this study focus is on the intermediate and high speed shafts and so the MB is not discussed in detail.
- 75 In Wang et al. (2020b) the MB is modelled within an overall numerical model of the drivetrain using SIMPACK software. The model consists of both rigid and flexible bodies, with bearings modelled as force elements with linear force-deflection relationships. High fidelity FE models of the critical components are developed in ANSYS before modal reduction is used to minimise degrees of freedom for reduced FE bodies in the system. The paper sought to determine 20-year drivetrain fatigue damage and found that the highest fatigue damage is experienced by the upwind MB. Wang et al. (2020a) determine MB
- 80 loading for the case where a flexible bedplate is included in modelling. Effects on damage equivalent fatigue loads are explored for flexible and rigid bedplate cases. The study concludes that flexibility in the bedplate leads to a reduction in loading and fatigue experienced by MBs when compared to the rigid case. Kock et al. (2019) use high fidelity FE models to investigate MB internal load distributions and contact pressures when considering variations in elasticity about the bearing circumference and clearance values. Their findings indicate that bearing housing elasticity strongly influences the number of rolling elements
- 85 under load and the maximum forces experienced by rolling elements.

A set of In addition to the analyses outlined above, work has also been undertaken in which simple drivetrain representations are used to study general characteristics of MB loads and their relationship to the incident wind field (Hart et al., 2019; Hart, 2020), with the current paper building directly on these. The first of these (Hart et al., 2019) considered load characteristics for different possible drivetrain configurations and demonstrated sensitivities to both wind field characteristics and drivetrain

90 setup. More recently, work was undertaken in which repeating structures in time-varying MB loading were identified and characterised, with impacts on the loading experienced by bearing rolling elements also studied. As touched upon in Section

1, the benefit of analytical models employed previously is their simplicity and speed, allowing large numbers of load cases to be analysed rapidly in order to seek possible identifiable trends or recurring off-design load events which may require more detailed scrutiny. While practical for such analyses, it is important to consider the accuracy of these models given their

95 inherent simplifying assumptions and the existence of different load reaction behaviours for different bearing types. These accuracy considerations form the focus of the current paper. The single MB model and turbulent wind field simulations from Hart et al. (2019) will be used here. As such, both will be described below in more detail.

Hart et al. (2019) performed a MB load analysis using simulated loading in realistic wind fields. The 3-dimensional turbulent wind fields were generated in DNV-GL's Bladed software using a Kaimal spectrum in accordance with IEC standards and six

- 100 different wind fields were created for every combination of the selected wind parameters as required for design certification (IEC, 2005)to describe the second order wind field statistics. The three parameters focused on were which characterise these wind fields are hub-height mean wind speeds (10, 12, 16, 20m/s), turbulence intensity (TL) (low, medium and high as specified by the IEC (2005)) and shear profile (power law shear exponents of 0.2 and 0.6)resulting in a total. 6 different wind fields were generated for each combination of these second order statistics using different initial random number seeds as required
- 105 for design certification (IEC, 2005). The above provides a total number of 144 wind profiles realistic 3D turbulent wind fields spanning a significant range of typical operational conditions. The 6 wind files fields associated with each combination of the parameters will be referred to as common parameter load sets (CPLS). Bladed was then used to run each wind file through fully aero-elastic multi-body simulations of a A 2MW wind turbine was then simulated operating in each of these 10 minute wind fields using DNV-GL Bladed aeroelastic software, with hub loading time series extracted. This resulted in 144 realistic
- 110 10 minute hub loading time series for the 2MW wind turbineand the, and it is these same load files which are used as inputs to models throughout the current paper. These hub loading time series extracted.

Simple engineering drawings were then applied to simplified models of MB set-ups (the one used in the current paper is outlined below) in order to study MB load characteristics. Drivetrain details were provided by Onyx Insightfor the study undertaken in Hart et al. (2019) which provide the dimensions of various MB set-ups and included the gearbox connections,

- 115 this included gearbox connections represented as radial and axial spring stiffness values linear springs. Three analytical models were then created including defined which included a single main-bearing (SMB) system and two double main-bearing (DMB) systems. The hub loading time series across the full range of wind files were then injected into the models and the bearing reaction forces extracted. The analytical model for the SMB drivetrain configuration is displayed shown in Figure 1 and will form the focus of this paper.
- 120 this is the case considered here.

The equation system for the SMB drivetrain set-up is statically determinate and can be solved by balancing the moments about the gearbox giving:

$$F = \frac{M + (L_1 + L_2)B}{L_2}.$$
(1)

Figure 1. 2D analytic Analytic model for single main-bearing set-up in one plane. The full model consists of two such representation, one in each of the horizontal and vertical planes (Hart et al., 2019).

Table 1. Parameters for all models.

| Model Parameters |                |  |  |  |
|-------------------------|----------------|--|--|--|
| $L_{ m L}$              | 2.145m  |  |  |  |
| $L_2$                   | 2.615m         |  |  |  |
| $K_{1}$                 | 8E07N/m |  |  |  |
| $K_2$                   | 4E06N/m |  |  |  |
| Ğ                       | 392280N        |  |  |  |
| Shaft Diameter          | 0.4m    |  |  |  |

125

It is important to note that the overall model consists of two of the type shown in Figure 1, with one in the horizontal and one in the vertical plane, with the resultant force being a vector combination of the two reaction forces at the MB. B and Mrepresent the force and moment loads at the hub  $\frac{1}{2}$  and  $L_1$  and  $L_2$  are 2.145m and 2.615m and represent the distances between the hub and MB, and MB and gearbox - respectively. The axial and radial springs to the right of the model ( $K_1$  and  $K_2$ ) represent the connection between the shaft and gearbox as stiffness values, while G represents the gearbox mass-weight in the vertical plane and is zero in the horizontal plane. F is the bearing support main-bearing reaction force. Findings demonstrated greatly varying mean and peak loads, as well as load ratios, between the different drivetrain configurations and high sensitivities 130 to wind field characteristics All model parameters can be found in Table 1.

While models and results in Hart et al. (2019) are promising demonstrate potentially important findings, the utilised models are simple, and hence come with limitations. The bearings are modelled as single point fixed supports, meaning all loads are reacted by a reaction force loading is reacted as forces at the MB - with no moment reaction reactions present. The two most

model also assumes the independence of loading and reaction behaviour in the horizontal and vertical planes. As outlined in 135 Section 1, two common bearings used for WT MBs are spherical roller bearings (SRB) which cannot react moment loadsand tapered roller bearings (TRB) which can react wind turbine MBs are DSRBs, which cannot support moment loads, and DTRBs, which can support both forces and moments (Yagi, 2004; Smalley, 2015; ?) (Yagi, 2004; Smalley, 2015; Hart et al., 2020). Therefore, the validity of existing models for-when representing different bearing types should and possible 3D effects is to be

140 considered. This validity is the focus of the current work.

**3 Comparison of Analytical and FEA modelsFinite Element Models**

In order to asses the effectiveness of the simple analytical models used thus far, two finite element (FE) FE models of the SMB system were created in ANSYS; with. The FE models were designed to be general and do not seek to represent any particular bearing specifically, but rather the global behaviour of different bearing types; one designed to behave like an SRB,

- a DSRB (non-moment reacting) and the other to behave like a TRB, as described below DTRB (does support moment loads). Likewise, the rest of the drivetrain system such as the shaft and gearbox connections remain both general and similar for the two different bearing types to create a like for like study. The models were subjected to the same hub loading as the analytical models, outlined in the previous section, with bearing support reaction forces outputted and compared with those from the analytical model. Both FE models share dimensions with the SMB analytical model. The FE models themselves still remain
   relatively simple, with relevant behaviours captured without the modelling of individual rolling elements as described below.
- To aid reproducibility a table of input and output value examples for all models is given in Table A1 of Appendix A.

SRB-DSRB FE Model - The SRB-DSRB FE model was created with 3 separate bodies; referred to here as the shaft, the bearing and the bearing housing - A fixed support was added to the base of the bearing housing to represent the connection to the bed plate and the connections between low speed shaft and gearbox was modelled by spring connections horizontally and vertically with stiffness values determined by Romax Technology software. (see Figure 3a). The bearing was connected to the body with a bonded connection and the bearing to bearing housing connection modelled as a deformable shaft using a bonded type contact and the convex outer face of the bearing was connected to the concave inner face of the bearing housing with a spherical joint. This type of connection allows the transfer of force loads from the shaft to the bearing and housing but will

- 160 not react moments, emulating SRB behaviour . A bearing housing to deformably react forces in the horizontal, vertical and axial axes while being able to move freely in the rotational degrees of freedom, allowing the non-moment reacting behaviour of a DSRB to be captured without the complex modelling of individual rollers. The full model is displayed in Figure 2 and a sliced view of the bearing, housing and shaft can be seen in Figure 3a side-by-side with SRB elements overlaid on the same image to demonstrate the interface type being represented. The mesh was sized to have larger element sizes across the
- 165 shaft with smaller elements around Bearing clearance is assumed to be zero since this parameter most directly influences the internal load distribution, rather than overall reaction force. The bedplate is assumed to be rigid in this model which, from previous work (Kock et al., 2019; Wang et al., 2020a), can be expected to provide conservatively higher bearing unit reaction force results than if bedplate flexibility were included1. A fixed support was added to the base of the bearing housing to represent the connection to the bed plate and the connection between the low speed shaft and the gearbox was modelled by
- 170 three body-to-ground spring connections in the horizontal, vertical and axial directions. Appropriate equivalent stiffness values

<sup>1This additional aspect of modelling will be considered in future work as progressively more complex representations are implemented.

of the low speed shaft to gearbox connections were determined with the use of Romax Technology software. The stiffness values, along with model dimensions, can be found in Table 1. The shaft, along with the rest of the model, was designed to be general and is modelled as a solid piece of material. Actual wind turbine main shafts tend to be a mostly solid piece of material, although a small hole will run throughout the centre to allow for wiring to pass to the hub. A sensitivity analysis was

175 therefore undertaken to determine the effect shaft thickness has on results, this can be found in Appendix B, findings indicate low sensitivity to this value. A convergence study was undertaken to determine appropriate mesh densities, resulting in smaller elements on the bearing and bearing housing to increase accuracy at the contact regions, housing bodies and larger elements on the shaft. Input hub loading was applied to the front face of the shaft, the gearbox weight was applied to the rear of the shaft in the vertical axis and main-bearing reaction forces extracted from the fixed support at the base of the housing.

Figure 2. (a) A split view of the SRB FE The 3-dimensional finite element model displaying the geometries of the with double row spherical roller bearing and housingtype reaction behaviour. (b) A split view of the TRB FE model displaying the geometries of the bearing and housing.

**Figure 3.** (a) A split view of the SRB FE model displaying the geometries of the bearing and housing. (b) A split view of the TRB FE model displaying the geometries of the bearing and housing. Note: The roller elements and mesh displayed in these images are for illustrative purposes only, a finer mesh was used for the simulations.

**TRB-DTRB FE Model** - The **TRB-DTRB** FE model was created with two separate bodies; referred to here as the shaft and the bearing/bearing housing (see Figure 3b). The bearing<del>and bearing housing were modeled</del>/bearing housing was modelled as one piece of material with a bonded connection and connected to the shaft using a bonded type contact. This assumes zero clearance between the rollers and housing (typically found in pre-loaded DTRBs) and allows the bearing unit to emulate the force and moment reaction properties of a TRB and the preloading of rollers. A cross section of the DTRB. The dimensions

- 185 of the model, assumptions of a rigid bedplate and fixed support connection from the base of the bearing/bearing housing to the bedplate are the same as that outlined above in the DSRB description. The low speed shaft equivalent connection to the gearbox and applications of hub and gearbox loading are also the same as described above. The full model is displayed in Figure 3b. The base of the 4 and a sliced view of the bearing/bearing housing and shaft can be seen in Figure 3b side-by-side with TRB elements overlaid on the same image to demonstrate the interface type being represented. Model parameters can be found in
- 190 Table 1. The shaft was again modelled as a solid piece of material. Sensitivity analysis results for this configuration, relating to shaft thickness, can also be found in Appendix B. A convergence study was again undertaken to determine appropriate mesh densities. The DTRB main-bearing housing was modelled with a fixed support to represent the connection to the main bed plate and the gearbox was again modelled by body-to-ground horizontal and vertical spring connections with the same stiffness properties as the SRB model-reaction forces were extracted from the fixed support at the base of the housing.

Figure 4. The 3-dimensional finite element model with double row tapered roller bearing type reaction behaviour.

- 195 **Bearing contact assumptions:** Internal contact conditions and load distributions around the bearing circumference are important (and non-linear) aspects of bearing behaviour. However, the SMB analytical model being studied is not designed to go to this level of detail instead outputting the reaction forces at (or equivalently the loads applied to) the MB. As such, the simplified FE representations for DSRB and DTRB bearings outlined above are considered reasonable for the following reasons: DSRB case DSRBs are self-aligning and hence provide force but not moment reactions across the bearing, as such,
- 200 the reaction force required to balance the system should remain the same irrespective of the spring properties, with only displacement magnitudes effected, since the system is determinate. DTRB case in the DTRB case the system supports moments through opposite force reactions over the two bearing rows in addition to providing an overall force reaction. Consequently, nonlinear contact properties of the rollers will influence the share between force and moment reactions at the MB. However, the non-linearity present in line contact rollers2 is only slight, with an exponent of 1.11 (Harris, 2006), and so
- 205 they are reasonably approximated as linear (Dowson and Higginson, 1977; Tibbits, 2005). Considering the research question posed in Section 1, it is therefore argued that the FE DTRB model presented here sensibly recreates load reaction behaviours of the desired type.

Plots of

<sup>2Including tapered and cylindrical cases.

**4 **Comparison of Analytical and Finite Element Models**

- 210 The analytical model presented in Section 2 was compared with the FE models described in Section 3 to determine its validity when the 2D orthogonality and simply-supported reaction assumptions are removed. The models were compared by performing a root mean squared error (RMSE) comparison results analysis between the reaction force results for the models across the whole range of turbulent wind field load time histories. Plots of RMSE between the analytical and two FE models are shown in Figures 5 and 7, along with example time series plots of the bearing unit MB reaction forces in Figures 6 and 8. The RMSE
- 215 plots present the mean and standard deviations from within each CPLS (which each capture results from 6 wind files with parameters in common) with respect to mean wind speed, turbulence intensity and shear profile. Note that mean wind speeds speed values are staggered for clarity.